# A *PTEN* variant uncouples longevity from impaired fitness in *Caenorhabditis elegans* with reduced insulin/IGF-1 signaling

Hae-Eun H. Park 1,4, Wooseon Hwang 2,4, Seokjin Ham1,4, Eunah Kim 1,4, Ozlem Altintas3, Sangsoon Park 1, Heehwa G. Son1, Yujin Lee1, Dongyeop Lee2, Won Do Heo 1 & Seung-Jae V. Lee 1✉

Insulin/IGF-1 signaling (IIS) regulates various physiological aspects in numerous species. In *Caenorhabditis elegans*, mutations in the *daf-2*/insulin/IGF-1 receptor dramatically increase lifespan and immunity, but generally impair motility, growth, and reproduction. Whether these pleiotropic effects can be dissociated at a specific step in insulin/IGF-1 signaling pathway remains unknown. Through performing a mutagenesis screen, we identified a missense mutation *daf-18(yh1)* that alters a cysteine to tyrosine in DAF-18/PTEN phosphatase, which maintained the long lifespan and enhanced immunity, while improving the reduced motility in adult *daf-2* mutants. We showed that the *daf-18(yh1)* mutation decreased the lipid phosphatase activity of DAF-18/PTEN, while retaining a partial protein tyrosine phosphatase activity. We found that *daf-18(yh1)* maintained the partial activity of DAF-16/FOXO but restricted the detrimental upregulation of SKN-1/NRF2, contributing to beneficial physiological traits in *daf-2* mutants. Our work provides important insights into how one evolutionarily conserved component, PTEN, can coordinate animal health and longevity.

[1] Department of Biological Sciences, Korea Advanced Institute of Science and Technology, Daejeon 34141, South Korea. [2] Department of Life Sciences, Pohang University of Science and Technology, Pohang 37673, South Korea. [3] School of Interdisciplinary Bioscience and Bioengineering, Pohang University of Science and Technology, Pohang 37673, South Korea. [4] These authors contributed equally: Hae-Eun H. Park, Wooseon Hwang, Seokjin Ham, Eunah Kim. ✉email: seungjaevlee@kaist.ac.kr

Aging is accompanied by a decline in biological functions and by pathophysiology of various age-associated diseases. To devise strategies for promoting long and healthy human lives, molecular mechanisms underlying the aging processes have been extensively investigated for the last several decades. Hundreds of mutations in aging-related genes have been identified in various model organisms[1]. However, genetic and environmental factors that have been shown to increase lifespan tend to also cause fitness defects. In addition, the lifespan of wild nematode strains negatively correlates with growth rates[2], a key developmental fitness parameter. Overall, biological strategies that promote longevity while simultaneously maintaining fitness are rare.

The insulin/insulin-like growth factor-1 (IGF-1) signaling (IIS) pathway is one of the most evolutionarily conserved aging-related pathways[1,3–7]. Mutations in *daf-2*, the sole *Caenorhabditis elegans* insulin/IGF-1 receptor gene, double lifespan[8]. *daf-2* mutants also exhibit enhanced resistance to various stresses, such as oxidative, heat, osmotic, and pathogenic stresses[5–7]. However, *daf-2* mutations also generally cause deleterious effects on health and fitness parameters[9], which have led to debate regarding the benefits of reducing IIS for healthy longevity[9–13]. At high temperatures (e.g. 25 °C) that do not impair the development of wild-type worms, *daf-2* mutants arrest at a hibernation-like developmental stage, the dauer larval stage[5,14]. In addition, *daf-2* reduction-of-function alleles lead to reduced developmental rate, brood size, and motility, all of which are important biological attributes for competitive fitness in nature.

In addition to *daf-2*, various aging-related factors in IIS pathways have been identified. These include *daf-18*, which encodes a worm ortholog of phosphatase and tensin homolog (PTEN) phosphatase, an enzyme that dephosphorylates phosphatidylinositol 3,4,5-trisphosphate ($PIP_3$) to phosphatidylinositol 4,5-bisphosphate ($PIP_2$)[15]. *daf-18(nr2037)*, a strong loss-of-function mutation [hereafter referred to as *daf-18(−)*], fully suppresses constitutive dauer formation and the long lifespan of *daf-2* mutants[16,17]. The DAF-16/FOXO transcription factor, which is activated downstream of DAF-18/PTEN, is required for both the constitutive dauer phenotype and the longevity of *daf-2* mutants[1,4,5]. Other transcription factors, including SKN-1/ nuclear factor erythroid 2-related factor 2 (NRF2) and heat shock transcription factor 1 (HSF-1), are also crucial for longevity and stress resistance in *daf-2* mutants[1,4–7,18]. These transcription factors regulate the expression of common and distinct subsets of target genes, which appear to elicit various physiological effects. However, it remains unclear whether specific IIS components and/or particular targets of these key transcription factors regulate discrete physiological aspects of IIS, including longevity, health span, and development.

In this study, we aimed to uncouple the increased lifespan and decreased fitness, including developmental defects and decreased adult functionality metrics, exhibited by *daf-2* mutant *C. elegans*. We performed a large-scale mutagenesis screen and found that a specific *daf-18* missense mutant allele, designated as *yh1*, fully suppressed slow development and partially rescued the reduced brood size observed in *daf-2* mutants with minimal decrease in the extended lifespan. We demonstrated that *daf-18(yh1)* mutation completely restored the reduced motility observed in young organisms and extended health span as measured by several physiological aspects in *daf-2* mutants. Through the analysis of global gene expression profiles, we showed that *daf-18(yh1)* allele was a weaker allele than the strong loss-of-function *daf-18(−)*. We found that *daf-18(yh1)* substantially decreased the lipid phosphatase activity of DAF-18/PTEN, while partly maintaining its protein phosphatase activity. Furthermore, we showed that *daf-18(yh1)* partially retained the activity of the DAF-16/FOXO in

*daf-2* mutants while preventing the adverse activation of the SKN-1/NRF2 that appears to underlie the reduction in lifespan and health span. These data indicate that the extent of DAF-18/ PTEN activity differentially affects various IIS-regulated physiological processes by calibrating the activities of these key longevity transcription factors. Our findings provide insights into strategies for healthy aging with less undesirable side effects by optimally modulating IIS.

## Results

**A genetic screen identified mutations that differentially affected pathogen resistance and developmental defects in *daf-2*/ insulin/IGF-1 receptor mutants.** We performed an ethyl methanesulfonate (EMS) mutagenesis screen to identify the suppressors of the constitutive dauer formation phenotype of *daf-2(e1370)* [*daf-2(−)*] mutants with minimal effects on resistance to the pathogenic bacteria, *Pseudomonas aeruginosa* (PA14) (Fig. 1a). We identified three such mutant alleles (*yh1*, *yh2*, and *yh3*), the penetrance of which in suppressing dauer phenotypes was complete in *daf-2* mutants at 25 °C, but also conferred resistance against PA14 above that in wild-type animals (Supplementary Fig. 1a–d). Upon sequencing, we found that the *yh1* allele bore a mutation that altered the evolutionarily conserved cysteine 150 residue in DAF-18/PTEN to tyrosine (Supplementary Fig. 1e–g and Supplementary Table 1). The *yh2* and *yh3* alleles resulted in premature termination and defective splicing, respectively, in the *daf-16/FOXO* (Supplementary Fig. 1h and Supplementary Table 1). These data are consistent with previous reports showing that genetic inhibition of *daf-18* or *daf-16* suppresses various phenotypes in *daf-2* mutants[1,4–7,16,17,19].

**A *daf-18/PTEN* mutation uncouples longevity and retarded development in *daf-2(−)* mutants.** We focused our analysis on *daf-18(yh1)*, as *daf-2(−); daf-18(yh1)* animals survived substantially longer than the other two mutants in response to PA14 infection (Supplementary Fig. 1b–d). An *mCherry*-fused *daf-18* transgene (*mCherry::daf-18*)[20] rescued dauer formation and pathogen resistance phenotypes of *daf-2(−); daf-18(yh1)* mutants (Fig. 1b, c). Therefore, *yh1* is a causative loss-of-function mutation in the *daf-18*. Moreover, we confirmed that outcrossed *daf-2(−); daf-18(yh1)* and *daf-2(−); daf-18(syb499)* [a CRISPR/Cas9 knock-in allele that contains the same mutation as *daf-18(yh1)*] animals exhibited increased survival on PA14 infection and enhanced clearance of PA14 compared with wild-type animals (Fig. 1d–g and Supplementary Fig. 6c). Contrarily, a strong loss-of-function mutation, *daf-18(−)*, completely suppressed the enhanced immunity of *daf-2(−)* mutants (Fig. 1d–g). Both *daf-18(yh1)* and *daf-18(syb499)* increased the rate of PA14 intake in *daf-2* mutants (Fig. 1h); therefore, enhanced resistance to PA14 infection caused by the C150Y change in DAF-18 protein appears to be independent of PA14 intake. Importantly, the *daf-18(yh1)* mutation only partly decreased the long lifespan of *daf-2(−)* mutants (Fig. 1i and Supplementary Fig. 2a–c). In addition, *daf-18(yh1)* in the *daf-2(−)* mutant background retained the enhanced resistance against oxidative and heat stresses compared with wild-type animals (Fig. 1j, k). In contrast, *daf-18(−)* largely suppressed these phenotypes (Fig. 1i–k and Supplementary Fig. 2a–c). Thus, *daf-2(−)* animals containing the *daf-18(yh1)* are long lived and resistant to pathogenic bacteria and abiotic stresses compared with wild-type animals.

We then tested the effects of *daf-18(yh1)* on lifespan and PA14 resistance conferred by other mutations in *daf-2*, a weak *daf-2(e1368)* allele and a strong ligand-binding domain-defective *daf-2(e979)* allele, by RNAi knockdown of *daf-2*[21,22], and by a hypomorphic *hx546* allele of *age-1*, which encodes

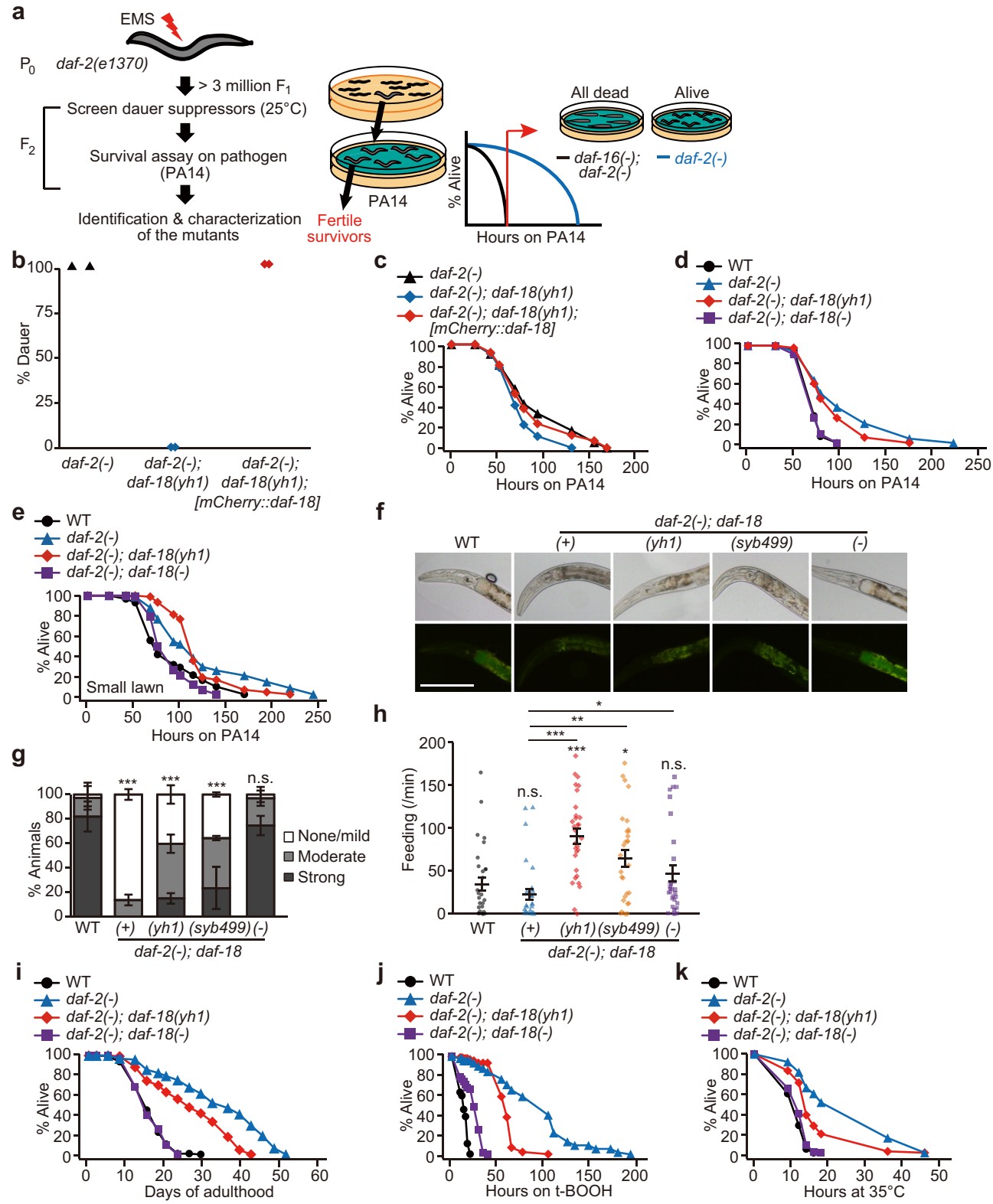

phosphoinositide 3-kinase that counteracts DAF-18/PTEN phosphatase[5,6]. We found that *daf-18(yh1)* partially maintained longevity and enhanced immunity in *daf-2(RNAi)* and *daf-2(e979)* worms (Supplementary Fig. 3a, b, e, f), but did not in *daf-2(e1368)* or *age-1(hx546)* mutants (Supplementary Fig. 3c, d, g, h). These results suggest that *daf-18(yh1)* can maintain lifespan and pathogen resistance in animals with reduced IIS caused by genetically inhibited *daf-2* with multiple intervention modes, not

specific to *daf-2(e1370)*. In addition, because *daf-2(e1368)* and *age-1(hx546)* mutations cause weak longevity and stress resistance phenotypes (Supplementary Fig. 3c, d, g–j)[21–25], we propose that a certain level of IIS reduction is required for *daf-18(yh1)* to maintain longevity and immunity.

**daf-18(yh1) ameliorates fitness defects caused by daf-2(−) mutations while maintaining overall health span.** Next, we

**Fig. 1 *daf-18(yh1)* suppresses developmental defects with minimal effects on longevity and resistance against various stresses in *daf-2(−)* mutants.**
**a** Schematic of forward genetic screen for fitness-regulating factors in *daf-2(e1370)* [*daf-2(−)*] mutants. Synchronized P₀ *daf-2(−)* animals were treated with ethyl methanesulfonate (EMS) to obtain > 3 million F₁ worms. The F₂ eggs were then transferred to OP50-seeded NGM plates and cultured at 25 °C for screening dauer suppressors. The dauer-suppressor mutants were immediately placed on plates seeded with pathogenic bacteria, *Pseudomonas aeruginosa* (PA14), for screening pathogen-resistant worms compared with *daf-16(mu86)* [*daf-16(−)*]; *daf-2(−)* mutants. When all the *daf-16(−); daf-2(−)* animals died but *daf-2(−)* animals were still alive, each of fertile survivors was singled and used for genetic analysis in this study (see Methods for details). **b**, **c** mCherry::daf-18 fully rescued dauer formation (*n* ≥ 433 for each condition, from two independent trials) (**b**) and the increased survival of worms on the pathogenic bacteria PA14 (*n* ≥ 180 for each condition, with big lawn where worms do not have space for an avoidance behavior) (**c**) in *daf-2(−); daf-18(yh1)* mutants to the levels of *daf-2(−)* animals. **d**, **e** Survival curves of outcrossed *daf-2(−); daf-18(yh1)* and *daf-2(−); daf-18(nr2037)* [*daf-18(−)*] mutants on PA14 big lawn (*n* = 180 for each condition) (**d**) and PA14 small lawn where worms can avoid the pathogen (*n* ≥ 180 for each condition) (**e**), compared with *daf-2(−)* and wild-type (WT) worms. **f** Representative images of indicated animals, WT, *daf-2(−); daf-18(+)*, *daf-2(−); daf-18(yh1)*, *daf-2(−); daf-18(syb499)*, and *daf-2(−); daf-18(−)*, after PA14-GFP exposure. Scale bar: 50 μm. **g** Quantification of PA14-GFP levels in the intestinal lumen of worms in (**f**) (*n* ≥ 25 for each condition, from three independent trials). Error bars represent the standard error of the mean (s.e.m., ****p* < 0.001, n.s.: not significant, chi-squared test relative to WT). **h** Feeding rate of the indicated strains after PA14-GFP infection (*n* = 30 for each condition, pharyngeal pumping per minute from three independent trials, **p* < 0.05, ***p* < 0.01, ****p* < 0.001, n.s.: not significant, two-tailed Student's *t* test relative to WT). **i–k** Lifespan (*n* ≥ 225 for each condition) (**i**), oxidative stress resistance (*n* = 180 for each condition) (**j**), and thermotolerance (*n* = 180 for each condition) (**k**) of WT, *daf-2(−)*, *daf-2(−); daf-18(yh1)*, and *daf-2(−); daf-18(−)* worms. See Supplementary Dataset 2, 3, and 4 for additional repeats and statistical analysis for the survival assay and fitness data shown in this figure. See also Source Data for data points used for the derivation of data.

assessed the developmental parameters and adult functionality metrics, including reproduction, motility, and feeding rates. We found that *daf-18(yh1)* significantly suppressed developmental defects in *daf-2(−)* mutants at a permissive temperature (20 °C) (Fig. 2a). *daf-18(yh1)* also rescued impaired reproduction and improved the reduced swimming (motility) and pumping (feeding) rates in young adult *daf-2(−)* mutants (Fig. 2b–d). In contrast, *daf-18(−)* did not increase the reduced swimming rates in young *daf-2(−)* adults (Fig. 2c). By measuring the age-dependent declines in motility and feeding rates, we found that the extended motility span in aging *daf-2(−)* mutants[9–12] was hardly decreased by *daf-18(yh1)* but was reduced by *daf-18(−)* (Fig. 2e–i). Together, these data indicate that *daf-18(yh1)* enhances the overall fitness parameters of young *daf-2(−)* mutants and improves the health span of these animals.

**daf-18(yh1) mutation decreases lipid phosphatase activity while retaining protein phosphatase activity of DAF-18/PTEN.** We next sought to determine whether *yh1* affected the PIP₃ phosphatase activity of DAF-18/PTEN. To this end, we generated transgenic animals that expressed the pleckstrin homology (PH) domain of mouse Akt fused with cyan fluorescent protein (CFP::PH_AKT), which bound plasma membrane-localized PIP₃ (Supplementary Fig. 4a–e)[26]. We found that both *daf-18(−)* and *daf-18(yh1)* increased the levels of plasma membrane-localized CFP::PH_AKT in *daf-2(−)* animals (Fig. 3a, b). Moreover, we found that the recombinant human PTEN protein harboring C105Y, the orthologous change caused by *C. elegans daf-18(yh1)*, exhibited substantially decreased lipid phosphatase activity (Fig. 3c–f); the effect was similar to that of the C124S change, which eliminated the lipid phosphatase activity (Fig. 3c–f)[27]. These data suggest that the C to Y change in DAF-18 and PTEN reduces the lipid phosphatase activity.

Because PTEN also acts as a protein tyrosine phosphatase[27,28], we performed protein phosphatase assays using a synthesized generic peptide harboring phospho-tyrosine. We found that PTEN^C105Y retained a substantial protein phosphatase activity (57.3%) compared with wild-type DAF-18/PTEN, whereas the phosphatase-dead PTEN^C124S (negative control) dramatically decreased that (Fig. 3g). We confirmed the results by measuring dose-dependent changes of the tyrosine phosphatase activity (Fig. 3h and Supplementary Fig. 4f). Thus, *daf-18(yh1)* appears to retain the partial tyrosine phosphatase activity of DAF-18/PTEN, raising the possibility that the protein phosphatase activity of

DAF-18^C150Y contributes to longevity and enhanced immunity in *daf-2(−)* animals.

**daf-18(yh1) is a hypomorphic allele that differentially affects dauer and pathogen resistance phenotypes in daf-2(−) animals.** Next, we analyzed transcriptional changes caused by *daf-18(yh1)* and *daf-18(−)* in *daf-2(−)* mutants by conducting mRNA-sequencing (RNA-seq) analysis. Principal component (PC) analysis demonstrated a separation of transcriptomes correlating with these different genotypes (Fig. 4a). We found that genes whose expression levels were altered by *daf-18(yh1)* in *daf-2(−)* mutants highly overlapped with those by *daf-18(−)* (fold change > 2, Benjamini and Hochberg (BH)-adjusted *p* < 0.05 out of 17,662 genes, Fig. 4b–f). Further RNA-seq analysis provided several lines of evidence suggesting that *daf-18(yh1)* was a weaker allele than *daf-18(−)*. First, transcriptomes of *daf-2(−); daf-18(yh1)* mutants were located between those of *daf-2(−)* and *daf-2(−); daf-18(−)* animals along a primary PC axis (PC 1) (Fig. 4a). Second, the number of genes whose expression was altered by *daf-18(yh1)* was smaller than those affected by *daf-18(−)* (Fig. 4c, d). Third, the overall magnitudes of gene expression changes caused by *daf-18(yh1)* were smaller than those caused by *daf-18(−)* (Fig. 4e, f and Supplementary Fig. 4g). Fourth, among genes that were upregulated in *daf-2(−)* mutants, the expression changes of 118 genes were greater compared with *daf-2(−); daf-18(−)* than with *daf-2(−); daf-18(yh1)* animals (fold change > 2, Fig. 4g–i, Supplementary Fig. 5a, and Supplementary Dataset 1). We also found that the expression changes of 219 genes downregulated in *daf-2(−)* mutants compared with *daf-2(−); daf-18(−)* were greater than those with *daf-2(−); daf-18(yh1)* mutants (fold change > 2, Fig. 4g–i, Supplementary Fig. 5b, and Supplementary Dataset 1). Thus, *daf-18(yh1)* appears to be a weaker hypomorphic allele than the strong loss-of-function *daf-18(−)* allele for transcriptomic changes in *daf-2(−)* mutants.

**daf-18(yh1) is a specific hypomorph that retains immunity and longevity.** We then tested the hypothesis that discrete thresholds exist for the suppression of different IIS-regulated phenotypes by a weaker allele, *daf-18(yh1)*, compared with a stronger allele, *daf-18(−)*. Specifically, *daf-2(−)* mutants may exhibit a lower threshold for the suppression of dauer formation than the enhanced pathogen resistance and longevity. In this scenario, similar to *daf-18(yh1)*, multiple hypomorphic mutations would fully suppress the dauer formation in *daf-2(−)* mutants while

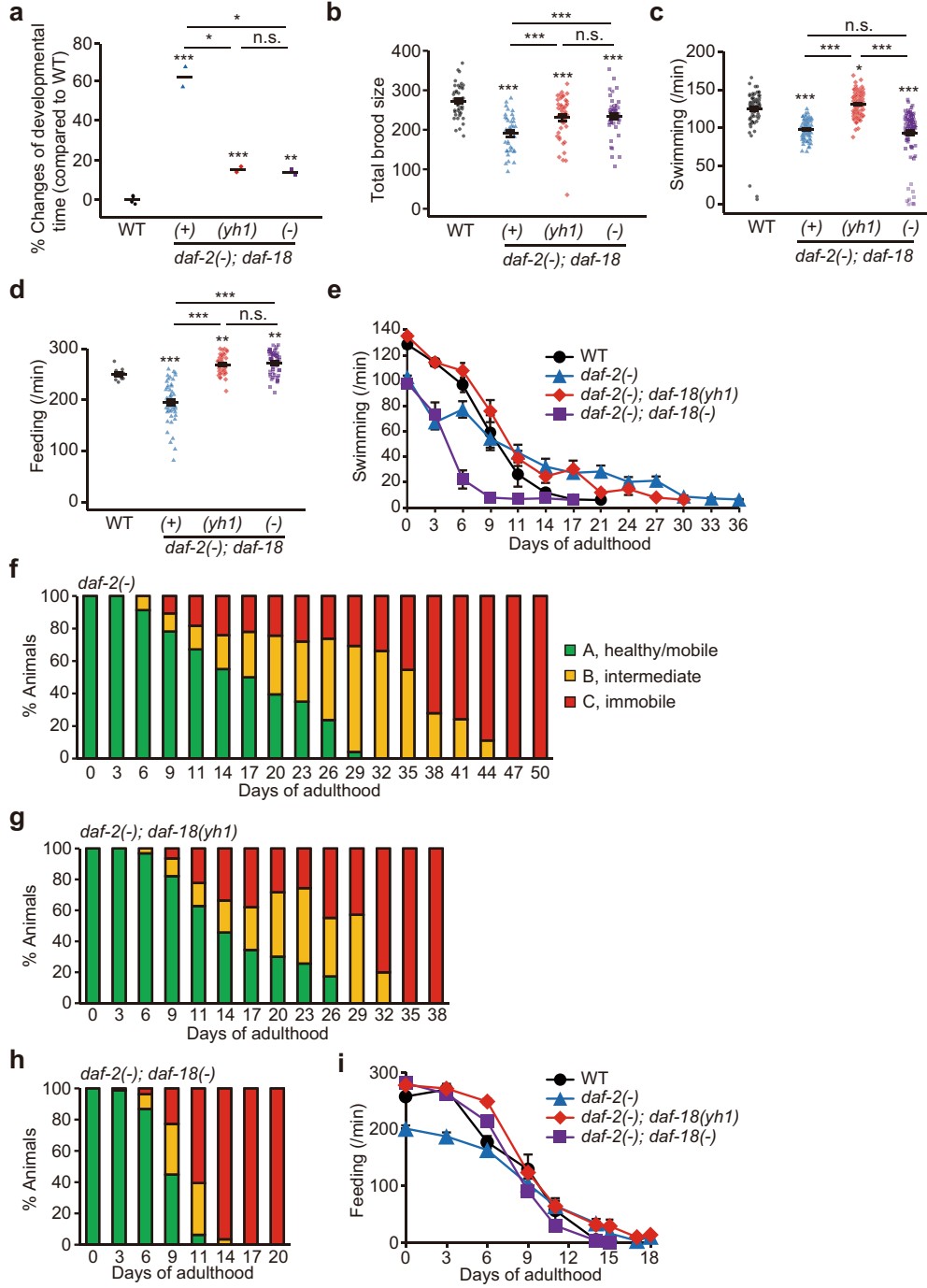

**Fig. 2 daf-18(yh1) improves health parameters in daf-2(−) mutants. a–d** Percent changes in developmental time ($n \geq 177$ for each condition, from two independent trials) (**a**), total brood size ($n \geq 37$ for each condition, from five independent trials) (**b**), swimming rate at day 0 ($n = 90$ for each condition, body bends per minute in liquid measured from nine independent trials) (**c**), and feeding rate at day 0 ($n = 50$ for each condition, pharyngeal pumping per minute measured from five independent trials) (**d**) of wild-type (WT), daf-2(e1370) [daf-2(−)]; daf-18(+), daf-2(−); daf-18(yh1), and daf-2(−); daf-18(nr2037) [daf-18(−)] animals. **e–i** Indicated health span of WT, daf-2(−), daf-2(−); daf-18(yh1), and daf-2(−); daf-18(−) animals. Shown are swimming span ($n = 10$ for each condition, from one trial) (**e**), motility span ($n = 20$ for each condition, from two independent trials) (**f–h**), and feeding span ($n = 10$ for each condition, from one trial) (**i**). Class A worms (green) are healthy and mobile, class B worms (yellow) display intermediate phenotypes, mobile but irregular movement, and class C worms (red) are immobile, in (**f–h**)[71]. See Supplementary Fig. 2f, g for the effects of daf-18(yh1) and daf-18(−) on the swimming span and feeding span in the WT background. Error bars indicate the standard error of the mean (s.e.m., *$p < 0.05$, **$p < 0.01$, ***$p < 0.001$, n.s.: not significant, two-tailed Student's $t$ test relative to WT unless otherwise noted). See Supplementary Dataset 4 for additional repeats and statistical analysis for the health span assay data shown in this figure. See also Source Data for data points used for the derivation of data.

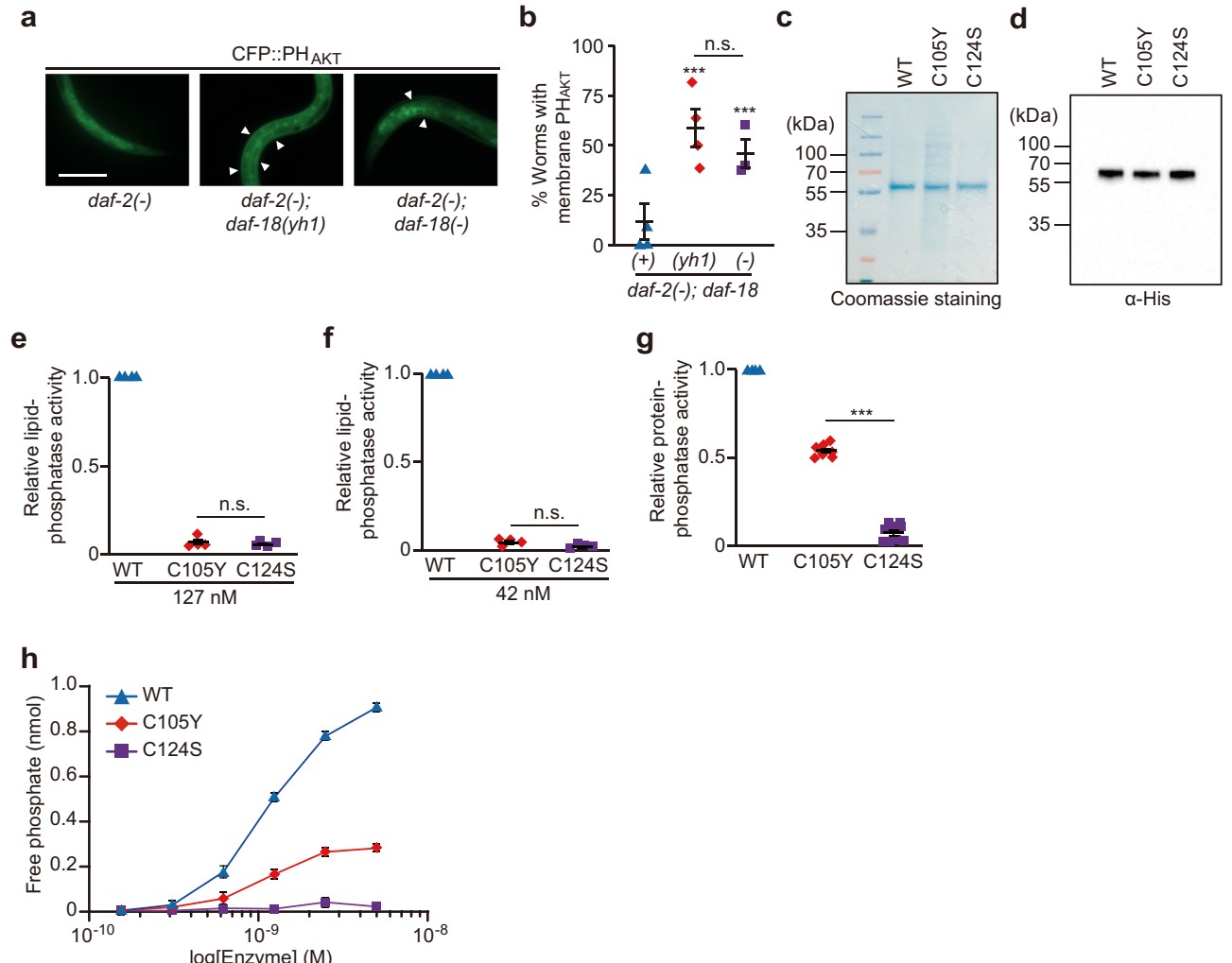

**Fig. 3 daf-18(yh1) substantially decreases lipid phosphatase activity, but partially maintains protein phosphatase activity of DAF-18/PTEN.**
**a** Representative images of *rpl-28p::CFP::PH_AKT*-expressing worms with *daf-2(e1370)* [*daf-2(−)*], *daf-2(−); daf-18(yh1)*, or *daf-2(−); daf-18(nr2037)* [*daf-18(−)*] mutations (*rpl-28p*: a promoter of a ubiquitous *rpl-28*, ribosomal protein large subunit 28). Scale bar: 50 μm. Arrowhead: membrane CFP::PH_AKT.
**b** Quantification of membrane-localized PH_AKT in worms in panel (**a**) ($n \geq 23$ for each condition, from four independent trials). **c, d** His-tagged human recombinant PTEN proteins used for in vitro phosphatase assay. WT: wild-type PTEN; C105Y: C105Y mutant PTEN; C124S: C124S mutant PTEN (phosphatase-dead variant). The recombinant proteins were separated by using SDS-PAGE and stained with Coomassie blue (**c**), and detected by using western blotting with anti-His antibody (**d**). **e, f** In vitro PTEN lipid phosphatase assay. Purified recombinant WT, C105Y, and C124S PTEN proteins [127 nM (**e**) or 42 nM (**f**)] were incubated with PIP_3 substrates ($N = 4$). **g** In vitro PTEN protein tyrosine phosphatase assay. Purified recombinant WT, C105Y, and C124S PTEN proteins were incubated with phospho-tyrosine-containing peptides, and the protein phosphatase activities were calculated by detecting free phosphates ($N = 9$). **h** Dose-dependent changes in protein phosphatase activities of the PTEN variants ($N = 3$). See Supplementary Fig. 4f for the comparison of protein phosphatase activities between WT PTEN and protein tyrosine phosphatase 1B (PTP1B), a positive control. Error bars represent the standard error of the mean (s.e.m., ***$p < 0.001$, n.s.: not significant, two-tailed Student's *t* test relative to WT unless otherwise noted). See Source Data for data points used for the derivation of data.

modestly affecting the increased pathogen resistance and longevity. By characterizing six additional *daf-18* mutant alleles (Supplementary Fig. 6a)[16,17,19,29–33], we found that five *daf-18* mutant alleles completely suppressed both dauer formation and PA14 resistance in *daf-2* mutants (Supplementary Fig. 6b, c). One exception was a missense mutant allele, *daf-18(pe407)*[33], which suppressed enhanced immunity in *daf-2* mutants while marginally decreasing dauer formation at 25 °C (Supplementary Fig. 6b, c). Due to the small number of tested *daf-18* mutant alleles, we cannot rule out different threshold levels for the suppression of *daf-2(−)* mutant phenotypes. However, these data are not consistent with the possibility that the threshold for the suppression of constitutive dauer phenotype in *daf-2(−)* mutants is lower than that of enhanced pathogen resistance. Thus, *yh1* appears to

be a specific reduction-of-function *daf-18* allele for retaining immunity and longevity while suppressing dauer formation in *daf-2(−)* mutants.

**daf-18(−) and daf-18(yh1) differentially affect downstream regulators of IIS.** We asked whether *daf-18(−)* and *daf-18(yh1)* differentially affected downstream regulators of IIS, which control various physiological processes in *daf-2(−)* animals. For this analysis, we compared our RNA-sequencing data to all the published transcriptome data that were obtained using animals with genetically inhibited *daf-2* (Fig. 5a, Supplementary Fig. 7, and Supplementary Table 2)[23,34–44]. We found that the expression of DAF-16/FOXO-induced genes was increased in *daf-2(−)* animals compared with *daf-2(−); daf-18(−)* and *daf-2(−); daf-*

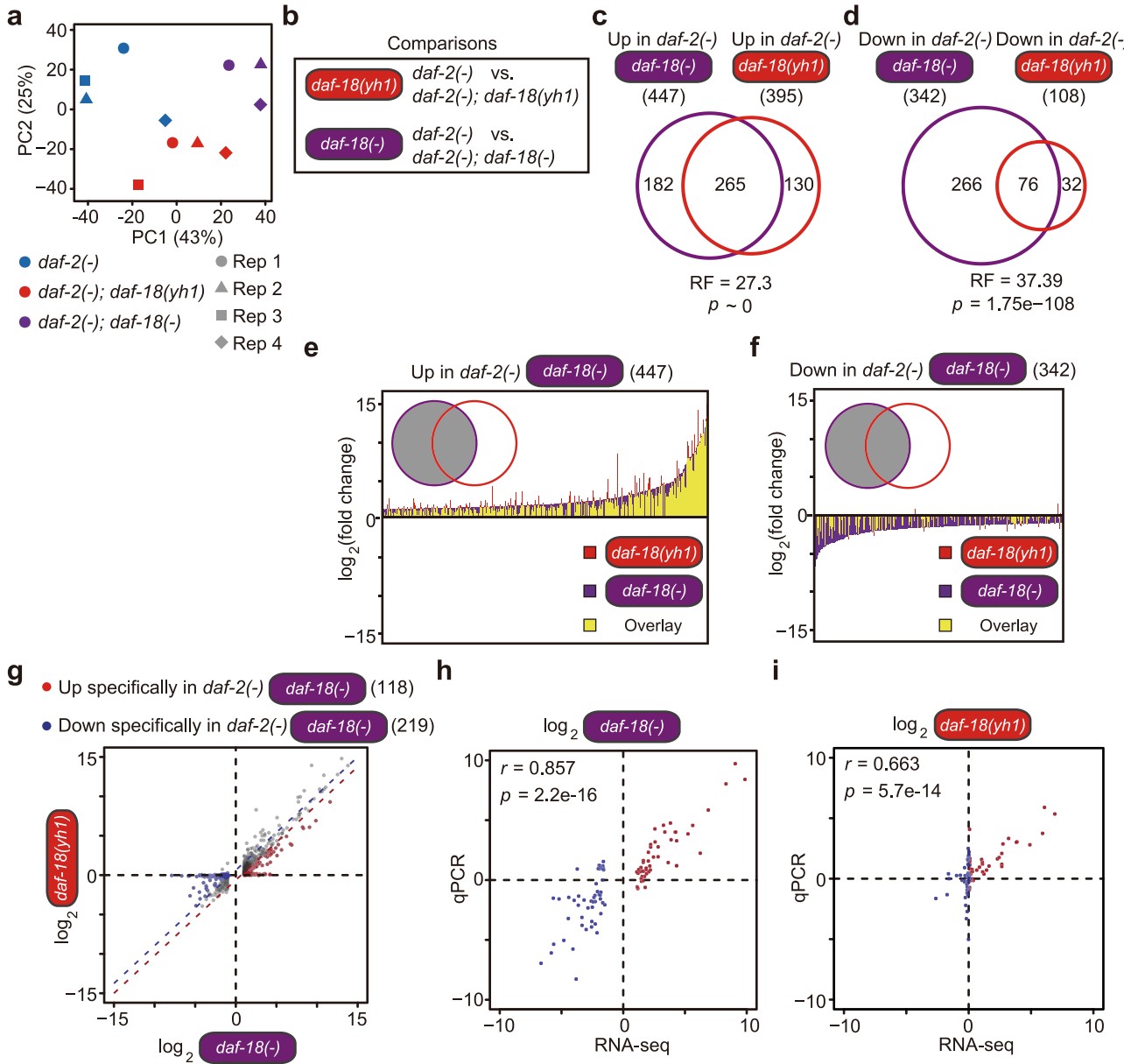

**Fig. 4 *daf-18(yh1)* is a hypomorphic allele that differentially affects gene expression in *daf-2(−)* mutants. a** A principal component (PC) analysis showing relative distance among samples. **b** Two comparisons for subsequent analyses of RNA-seq data [red: *daf-2(−)* vs. *daf-2(−); daf-18(yh1)*, purple: *daf-2(−)* vs. *daf-2(−); daf-18(−)*]. **c, d** Overlaps between genes upregulated (**c**) and downregulated (**d**) in *daf-2(−)* mutants compared to *daf-2(−); daf-18(−)* or *daf-2(−); daf-18(yh1)* mutants (*p* values were calculated by using exact hypergeometric probability test. RF: representation factor). **e, f** Comparisons of the extent of gene expression changes between the two comparisons. Genes whose expression was upregulated (**e**) and downregulated (**f**) in *daf-2(−)* mutants compared to *daf-2(−); daf-18(−)* mutants were considered. Overlaps between purple and red bars were marked in yellow. **g** A scatter plot showing the effects of *daf-18(−)* and *daf-18(yh1)* mutations on gene expression in *daf-2(−)* mutants. Shown are genes whose expression levels were specifically upregulated (red dot: 118 genes) or downregulated (blue dot: 219 genes) in *daf-2(−)* animals compared with *daf-2(−); daf-18(−)* worms (fold change > 2, Benjamini and Hochberg (BH)-adjusted *p* value < 0.05), but were only slightly altered compared with *daf-2(−); daf-18(yh1)* animals (fold change < 2 relative to *daf-18(−)*) in *daf-2(−)* mutants. See Supplementary Fig. 4 h, i, l, m for gene ontology and tissue enrichment analysis results of these genes. **h, i** Confirmation of the expression changes of the top 50-ranked genes in (**g**) by using quantitative RT-PCR. Both axes are log₂-transformed. See Supplementary Fig. 5 and Supplementary Dataset 1 for details.

*18(yh1)* animals, but the impact of *daf-18(−)* was stronger than that of *daf-18(yh1)* (Fig. 5a and Supplementary Fig. 7b). Consistently, *daf-18(yh1)* affected nuclear localization of DAF-16::GFP in *daf-2(−)* mutants at an intermediate level between *daf-18(+)* and *daf-18(−)* (Fig. 5f, g and Supplementary Fig. 8). Thus, *daf-18(yh1)* appears to retain the partial activity of the DAF-16/FOXO transcription factor. In contrast, we showed that the effects of *daf-18(yh1)* on the expression levels of DAF-16/

FOXO-repressed genes in *daf-2(−)* mutants were similar to those of *daf-18(−)* (Supplementary Fig. 7a, c). These results raise a possibility that factors other than DAF-16/FOXO underlie the differences in gene expression caused by *daf-18(−)* and *daf-18(yh1)* in *daf-2(−)* mutants.

Among such candidate factors, the expression of SKN-1/NRF2-induced genes was increased by *daf-18(−)* in *daf-2(−)* backgrounds while not being substantially affected by *daf-18(yh1)*

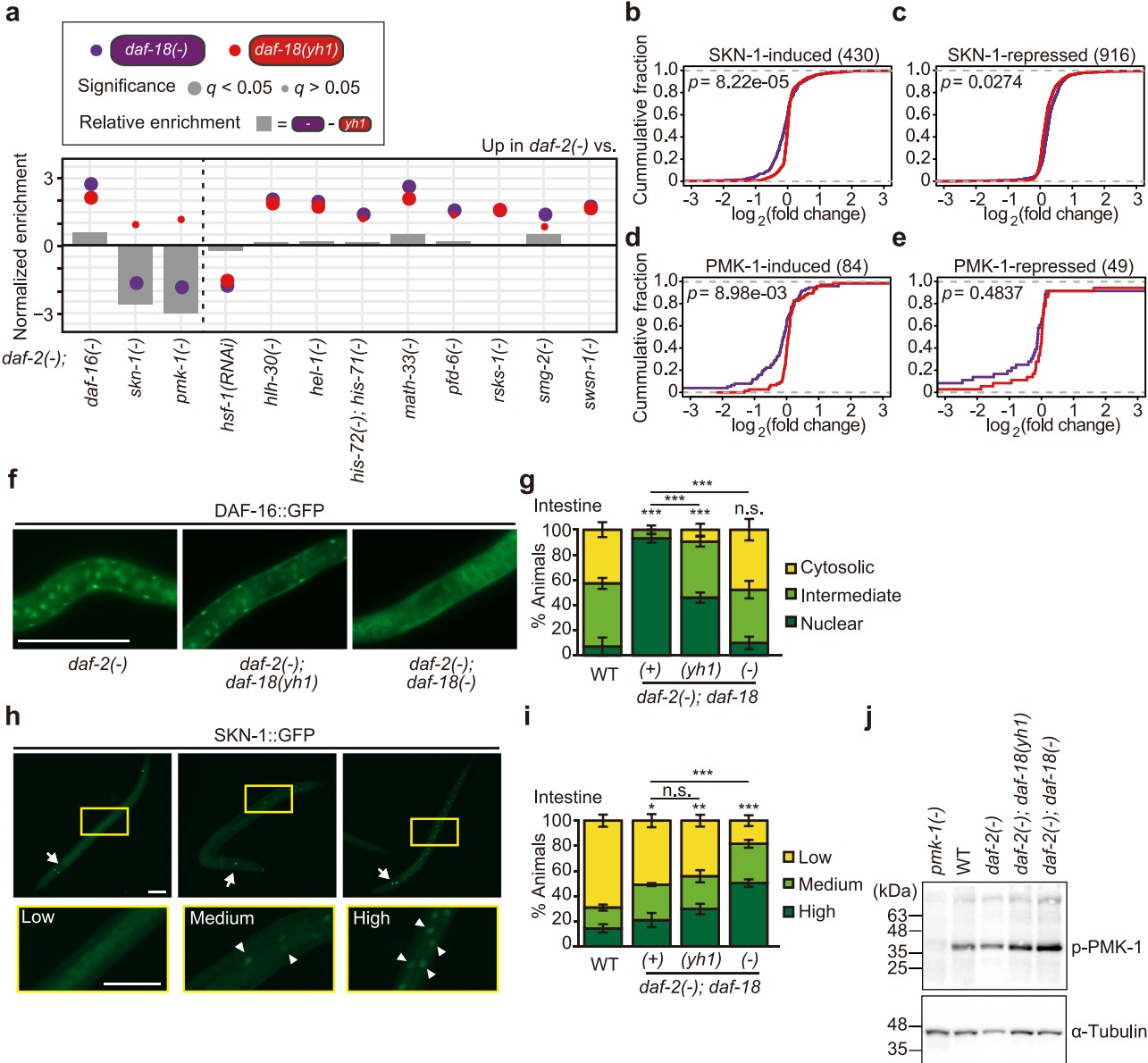

(Fig. 5a, b and Supplementary Fig. 9a, b, e, g, i). We also obtained similar results for target genes upregulated by PMK-1/p38 MAP kinase (Fig. 5a, d), which acts upstream of the SKN-1/NRF2 transcription factor[18,45] (See Fig. 5c, e for the effects of *daf-18(−)* and *daf-18(yh1)* on the expression of SKN-1/NRF2- and PMK-1/ p38 MAP kinase-repressed genes in the *daf-2(−)* background). Thus, the strong *daf-18(−)* mutant allele appears to upregulate PMK-1/p38 MAP kinase and SKN-1/NRF2 signaling in *daf-2(−)* animals. Consistently, we found that *daf-18(−)* significantly increased the nuclear localization of SKN-1::GFP in *daf-2(−)* mutants, but *daf-18(yh1)* had a smaller impact (Fig. 5h, i). We also found that the level of active, phospho-PMK-1 in *daf-2(−)* mutants was increased by *daf-18(−)*, while not being substantially affected by *daf-18(yh1)* (Fig. 5j). Thus, *daf-18(yh1)* appears to have smaller effects on the activity of PMK-1 than *daf-18(−)*. Moreover, the expression of genes downregulated in *daf-2(−)* mutants, which was lower in *daf-2(−); daf-18(yh1)* than in *daf-2(−); daf-18(−)* animals, was generally decreased in *daf-2(−); skn-1(−)* mutants (Supplementary Fig. 9a, b). Thus, SKN-1/ NRF2-induced gene expression was highly upregulated in *daf-2(−); daf-18(−)* mutants compared with *daf-2(−); daf-18(yh1)*

animals. Contrary to DAF-16/FOXO and SKN-1/NRF2, *daf-18(yh1)* and *daf-18(−)* similarly affected the target gene expression of various other analyzed signaling factors similarly in *daf-2* mutants, including HSF-1 and HLH-30/TFEB (Fig. 5a and Supplementary Fig. 7a; see also Fig. 5 legends for more details). These data suggest that *daf-18(−)* and *daf-18(yh1)* differentially affect specific downstream regulators of IIS, including transcription factors DAF-16/FOXO and SKN-1/NRF2.

We further examined the effects of the six additional *daf-18* alleles that we physiologically tested (Supplementary Fig. 6a–c) on the activities of DAF-16/FOXO and SKN-1/NRF2 by measuring their subcellular localization. We found that three *daf-18* alleles, *e1375*, *mu398*, and *pe407*, retained the DAF-16::GFP nuclear localization with an extent similar to or higher than that of *yh1* in *daf-2* mutants (Supplementary Fig. 6d). We then showed that all these six *daf-18* alleles increased the nuclear localization of SKN-1::GFP in *daf-2(e1370)* animals different from *daf-18(yh1)* (Supplementary Fig. 6e). These results indicate that *daf-18(yh1)* is a distinctive allele that did not hyperactive SKN-1/NRF2, unlike all the other tested *daf-18* alleles. Together, these results suggest that *yh1* is a specific *daf-18* hypomorphic allele that confers

**Fig. 5 Differential effects of *daf-18(yh1)* and *daf-18(−)* on downstream factors of IIS. a** Normalized enrichment of expression changes of indicated genes in *daf-2(e1370)* [*daf-2(−)*] mutants compared with *daf-2(−); daf-18(yh1)* and with *daf-2(−); daf-18(nr2037)* [*daf-18(−)*] mutants. DAF-16/FOXO[34], SKN-1/NRF2[23], PMK-1/p38 MAP kinase[35], HSF-1/heat shock factor 1[36], HLH-30/TFEB[37], HEL-1/DEAD-box RNA helicase[38], histone H3.3[39], MATH-33/deubiquitylating enzyme (at 25 °C)[40], PFD-6/prefoldin 6[41], RSKS-1/S6K[42], SMG-2/UPF1[43], and SWSN-1/BAF155/170[44] target genes are shown (See Supplementary Table 2 for details). MATH-33 upregulates DAF-16/FOXO as its deubiquitylating enzyme[40], and therefore the data with MATH-33 display a tendency similar to those with DAF-16/FOXO. Relative enrichment indicates the difference of gene expression changes caused by *daf-18(−)* and by *daf-18(yh1)* in *daf-2(−)* worms. *q* values were obtained by calculating the false discovery rate corresponding to each normalized enrichment. **b–e** Cumulative fraction of genes in an ascending order of the extent of gene expression changes conferred by *daf-18(yh1)* and *daf-18(−)* in *daf-2(−)* animals. **b, c** Shown are genes whose expression was upregulated (**b**) and downregulated (**c**) in *daf-2(−)* and *daf-2(e1368)* worms compared to *skn-1(zu67)* mutants[23]. **d, e** Genes whose expression was upregulated (**d**) and downregulated (**e**) in *daf-2(e1368)* worms compared to *pmk-1(km25)* mutants are shown[35]. Corresponding gene set enrichment analysis and calculation of cumulative fractions of other SKN-1/NRF2-induced or repressed genes are shown in Supplementary Fig. 9. **f–i** Effects of *daf-18* mutations on subcellular localization of DAF-16::GFP and SKN-1::GFP in *daf-2(−)* mutants. **f** Representative images of the subcellular localization of DAF-16::GFP in the intestines of *daf-2(−)*, *daf-2(−); daf-18(yh1)*, and *daf-2(−); daf-18(−)* animals. Scale bar: 50 μm. **g** Quantification of data shown in (**f**) in addition to the subcellular localization of DAF-16::GFP in wild-type (WT). Cytosolic: predominant cytosolic localization, intermediate: partial nuclear localization, nuclear: predominant nuclear localization (*n* ≥ 32 for each condition, from four to seven independent trials). See Supplementary Fig. 8 for the subcellular localization of DAF-16::GFP in neurons, intestine, and hypodermis. **h** Representative images of worms expressing SKN-1::GFP at low, medium, or high levels in the nuclei of intestinal cells. Scale bar: 50 μm. Arrow: ASI neurons. Arrowhead: nuclear SKN-1::GFP. Yellow boxes indicate magnified trunk regions of animals expressing SKN-1::GFP. **i** Quantification of the nuclear localization of SKN-1::GFP in the intestinal cells of indicated strains. Low: very dim GFP in the nuclei, medium: < 50% of the nuclei with SKN-1::GFP, high: > 50% of the nuclei with SKN-1::GFP (*n* ≥ 291 for each condition, from eight independent trials). Error bars represent the standard error of the mean (s.e.m., *$p$ < 0.05, **$p$ < 0.01, ***$p$ < 0.001, chi-squared test relative to WT unless otherwise noted). **j** Phospho-PMK-1 detection in WT, *daf-2(−)*, *daf-2(−); daf-18(yh1)*, and *daf-2(−); daf-18(−)* by using western blot assay (*N* = 5). *pmk-1(km25)* [*pmk-1(−)*] animals were used for the antibody validation (*N* = 2). α-tubulin was used as a loading control. See Source Data for data points used for the derivation of data.

beneficial physiological traits in *daf-2* mutants by limiting the hyperactivation of SKN-1/NRF2 and retaining partial activity of DAF-16/FOXO in *daf-2(−)* mutants.

***daf-18(yh1)* tunes the activities of DAF-16/FOXO and SKN-1/NRF2 to maintain healthy longevity.** Next, we functionally tested the differential effects of *daf-18(yh1)* and *daf-18(−)* on the activities of DAF-16/FOXO and SKN-1/NRF2 by measuring lifespan and health parameters. We found that RNAi knockdown of *daf-16* robustly decreased the longevity of *daf-2(−)* and *daf-2(−); daf-18(yh1)* animals while marginally shortening that of *daf-2(−); daf-18(−)* animals (Fig. 6a and Supplementary Fig. 10a). These data suggest that the activity of DAF-16/FOXO contributes to the longevity retained in *daf-2(−); daf-18(yh1)* animals. We then found that a *skn-1* gain-of-function mutation [*skn-1(gf)*][46] decreased the extended lifespan of *daf-2(−)* and *daf-2(−); daf-18(yh1)* worms, but not that of *daf-2(−); daf-18(−)* worms (Fig. 6b and Supplementary Fig. 10b). Conversely, we found that a reduction-of-function allele *skn-1(zj15)* [*skn-1(−)*][47] extended the short lifespan of *daf-2(−); daf-18(−)* animals, but not that of *daf-2(−); daf-18(yh1)* animals (Fig. 6c). In addition, we showed that knockdown of *pmk-1*, which acts upstream of SKN-1/NRF2, increased the lifespan of *daf-2(−); daf-18(−)* worms, but not that of *daf-2(−)* or *daf-2(−); daf-18(yh1)* worms (Fig. 6d and Supplementary Fig. 10d). These data suggest that hyperactivation of SKN-1/NRF2 contributes to the short lifespan of *daf-2(−); daf-18(−)* worms. Additionally, the effects of *daf-16(mu86)* [*daf-16(−)*], *skn-1(gf)*, and *pmk-1* RNAi on the age-dependent motility decrease in *daf-2(−); daf-18(yh1)* and *daf-2(−); daf-18(−)* animals mirrored the effects on lifespan (Fig. 6e, f, h). Although *skn-1(−)* did not affect the motility of *daf-2(−); daf-18(−)* or *daf-2(−); daf-18(yh1)* animals (Fig. 6g; see figure legends for more information), these data indicate that modulating the activity of DAF-16/FOXO and SKN-1/NRF at a proper level can enhance health span in *daf-2(−)* mutants. Altogether, *daf-18(yh1)* appears to confer healthy longevity with minimal detrimental effects on the fitness of animals with reduced IIS by retaining DAF-16/FOXO activity while simultaneously dampening the harmful activation of SKN-1/NRF2 (Fig. 6i).

## Discussion

Uncoupling the association between longevity and reduced fitness, including developmental defects and decreased adult functionality metrics, has been a major challenge in the field of aging research. In this report, we identified a specific missense mutation in the *daf-18/PTEN* that sustained the long lifespan and enhanced immunity conferred in *daf-2*/insulin/IGF-1 receptor mutant *C. elegans*, without apparent accompanying defects in development and health span. Notably, our data revealed that a specific mutation in *daf-18/PTEN* preserved the partial protein phosphatase activity of DAF-18/PTEN and transcriptional activity of DAF-16/FOXO while preventing the harmful activation of transcription factor SKN-1/NRF2, leading to the differential physiological effects. Thus, a proper balance between DAF-16/FOXO and SKN-1/NRF2 activities appears to promote health span in animals with reduced IIS. These data indicate that the modulation of DAF-18/PTEN activity differentially regulates DAF-16/FOXO and SKN-1/NRF2 in reduced IIS and, in turn, uncouples various pleiotropic phenotypes caused by reduced IIS.

Recent studies have drawn controversy over the effects of *daf-2* mutations on health span despite the consensus of the effects of these mutations on extreme longevity[9–13]. For example, *daf-2* mutations have been reported to increase lifespan by mostly prolonging the unhealthy period of old age[9], through decreasing gut colonization by dietary bacteria[11]. Contrarily, we previously revealed that *daf-2* mutants exhibit extended healthy periods throughout aging by measuring a maximum physical ability[10] and that temporal inhibition of *daf-2* enhances immunocompetence in old age[36]. All these studies used genetic inhibition of the *daf-2*, mutant alleles and RNAi, to measure multiple aspects of health span, without mutations in other genes[9–13]. Here, we aimed to modulate the activity of additional components of IIS to improve the health span in *daf-2* mutant worms. Our data demonstrated that the specific change in DAF-18/PTEN caused by *yh1* increased the fitness and health span in *daf-2* mutants with minimal unfavorable effect on lifespan. Thus, DAF-18/PTEN can be used as a calibrator for achieving healthy longevity.

Our data indicate that *daf-18(yh1)* retains nuclear localization of DAF-16/FOXO in *daf-2(−)* mutants (Fig. 5f, g), while downregulating DAF-16/FOXO-induced genes in *daf-2(−)* mutants

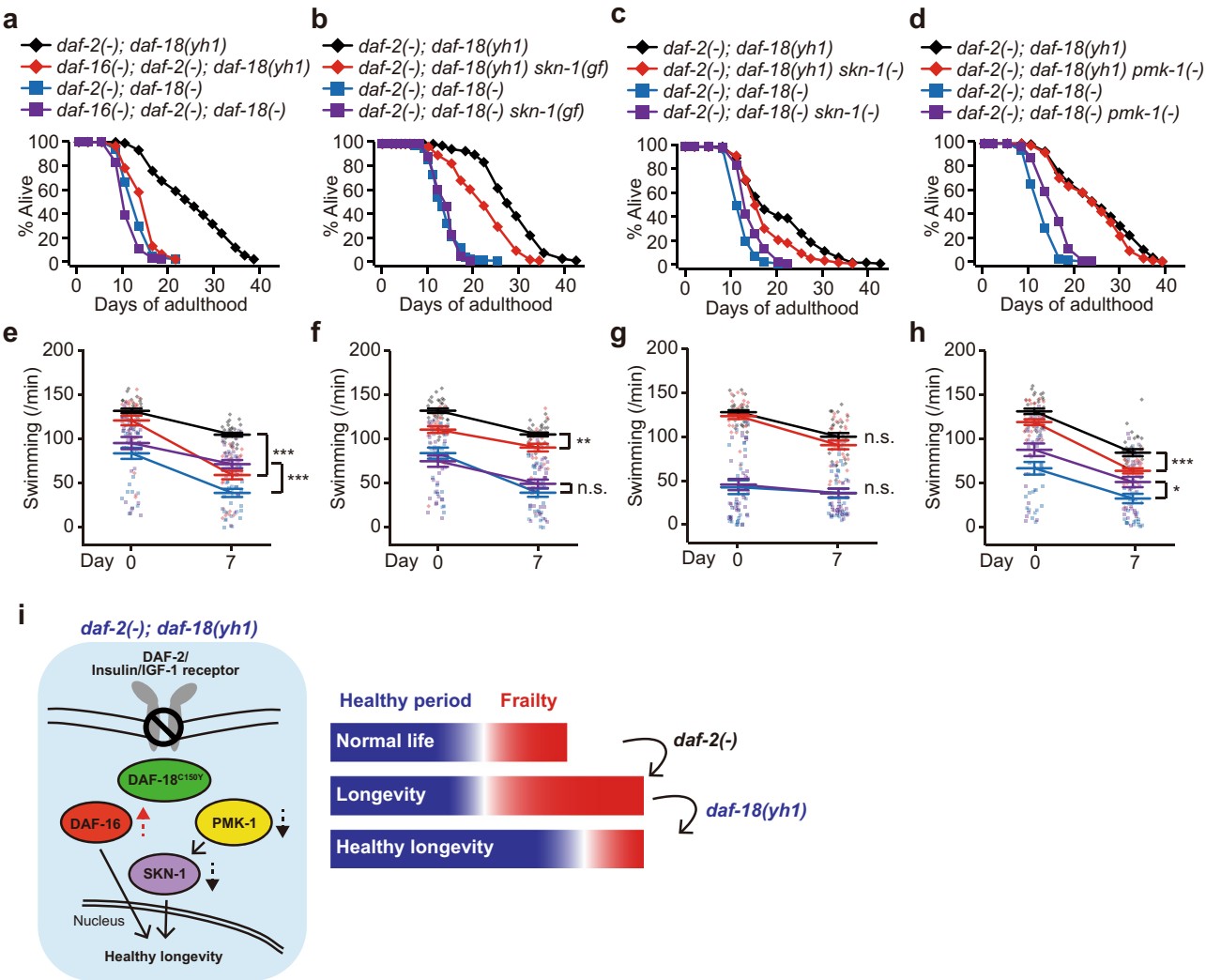

**Fig. 6 *daf-18(yh1)* exerts healthy longevity in *daf-2(−)* animals by maintaining the activity of DAF-16/FOXO while reducing that of SKN-1/NRF2.**
**a–d** Effects of *daf-16(RNAi)* [*daf-16(−)*] (**a**), *skn-1(lax188)* [*skn-1(gf)*] (**b**), *skn-1(zj15)* [*skn-1(−)*] (**c**), and *pmk-1(RNAi)* [*pmk-1(−)*] (**d**) on the lifespan of *daf-2(e1370); daf-18(yh1)* [*daf-2(−); daf-18(yh1)*] and *daf-2(−); daf-18(nr2037)* [*daf-18(−)*] animals (*n* ≥ 240 for each condition). All the lifespan assays were performed at least twice independently. **e–h** Effects of *daf-16(mu86)* (**e**), *skn-1(gf)* (**f**), *skn-1(−)* (**g**), and *pmk-1(−)* (**h**) on the swimming rate (motility) of *daf-2(−); daf-18(yh1)* and *daf-2(−); daf-18(−)* animals at day 0 and day 7 adulthoods (*n* = 30 for each condition, from three independent trials). See Supplementary Fig. 10c for the effects of *skn-1(−)* on the lifespan of *daf-2(−)* and wild-type (WT) animals. See Supplementary Fig. 10e, f for the requirement of PMK-1 for the decreased PA14 susceptibility of *daf-2(−)* and *daf-2(−); daf-18(yh1)* worms and for the normal survival of WT on PA14. We found that *skn-1(−)* did not affect either the motility of *daf-2(−); daf-18(−)* or that of *daf-2(−); daf-18(yh1)* worms in (**g**), and this may be due to the weak nature of the *zj15* allele, which needs to be tested in various other genetic backgrounds in future research. Error bars represent the standard error of the mean (s.e.m., *$p < 0.05$, **$p < 0.01$, ***$p < 0.001$, n.s.: not significant, two-tailed Student's *t* test). **i** A schematic showing that *daf-18(yh1)* retains partial transcriptional activity of DAF-16/FOXO while suppressing the hyperactivation of PMK-1/p38 MAPK and SKN-1/NRF2 in *daf-2(−)* mutant animals, which contributes to healthy longevity. Thus, *daf-18(yh1)* increases healthy periods, while decreasing the time of frailty caused by longevity-promoting *daf-2(−)* mutations. A dotted, upward arrow indicates partial activity maintenance, and dotted, downward arrows indicate suppression of hyperactivation. See Supplementary Dataset 2 and 4 for additional repeats and statistical analysis for the lifespan and health span assay data shown in this figure. See also Source Data for data points used for the derivation of data.

similarly to *daf-18(−)* (Fig. 5a). Thus, the nuclear localization status of DAF-16 does not appear to always correlate with target gene expression levels. One possible interpretation for this is that target gene expression of DAF-16/FOXO is affected by the surrounding density of cofactors or inhibitors of the DAF-16/FOXO. In addition, DAF-16-binding elements (DBEs) associated with direct targets of DAF-16/FOXO and DAF-16-associated elements (DAEs) associated with indirect targets may be differentially regulated by *daf-18(−)* and *daf-18(yh1)* in *daf-2(−)* mutants. Further research is required to test these possibilities for better understanding of regulation of DAF-16/FOXO by DAF-18/PTEN.

In mammalian cells and tissues, PTEN disruption enhances the transcriptional activity of NRF[48–51]. For example, 80% of *PTEN*-negative human patients with endometrioid carcinomas exhibit increased expression of *NRF2* and its targets[49]. Mammalian PTEN also decreases NRF protein levels via ubiquitin-mediated proteolysis[51]. Paradoxically, the genetic inhibition of the *C. elegans* DAF-2/insulin/IGF-1 receptor, which increases the level of DAF-18/PTEN[52], increases the activity of SKN-1/NRF2 in *C. elegans*[18,23,53]. However, these previous studies using *C. elegans* did not directly test the causal role of DAF-18/PTEN in the regulation of SKN-1/NRF2 activity. Our current study indicates

that strong genetic inhibition of DAF-18/PTEN increases SKN-1/NRF2 activity in *C. elegans* with reduced IIS, which contributes to shortened lifespan and health span, consistent with the studies on mammals. In addition, recent studies suggest that hyperactivation of SKN-1/NRF2 can impair worm health[46,54]. Thus, we propose that DAF-18/PTEN acts as a negative regulator of SKN-1/NRF2 in *C. elegans* with reduced IIS and contributes to the modulation of health span.

Mutations in *PTEN*, a tumor suppressor, underlie the pathology of various human cancers[55,56]. A human mutation that results in PTEN$^{C105Y}$, which corresponds to DAF-18$^{C150Y}$ in *C. elegans daf-18(yh1)*, leads to an autosomal dominant syndrome, Bannayan–Riley–Ruvalcaba syndrome[57]. This disorder is characterized by hamartomatous polyps in the intestine and benign subendothelial lipomas[58]. Here, we showed that the *daf-18(yh1)* mutation reduced the lifespan and health span in wild-type animals (Supplementary Fig. 2a-c, f, g) but maintained extended lifespan, health span, enhanced stress resistances, and overall fitness in animals with genetically inhibited *daf-2* (Fig. 1 and Fig. 2). These findings raise a possibility that IIS reduction in mammals bearing mutations in *PTEN* may improve the fitness and/or extend the health span of these animals. It is noteworthy that both activation and repression of NRF2 is implicated in the progression and development of tumors[59]. Our current work also suggests that various alleles in *daf-18/PTEN* can differentially affect the transcriptional activity of SKN-1/NRF2. In conclusion, it will be necessary to properly modulate the activity of PTEN and NRF to develop therapeutic strategies for treating human cancer patients.

## Methods

**Strains**. *C. elegans* strains were maintained at 15 °C or 20 °C on standard nematode growth medium (NGM) plates seeded with *E. coli* OP50 bacteria. Strains that were used in this study were outcrossed at least four times to wild-type N2 strain if not stated otherwise. The *C. elegans* strains used in this study are as follows: wild-type Bristol N2, CF1041 *daf-2(e1370) III*, CF1085 *daf-16(mu86) I; daf-2(e1370) III*, CF1042 *daf-16(mu86) I*, IJ713 *daf-2(e1370) III; daf-18(yh1) IV* (unoutcrossed), IJ1592 *daf-2(e1370) III; daf-18(yh1) IV*, IJ1417 *daf-16(yh2) I; daf-2(e1370) III* (unoutcrossed), IJ1418 *daf-16(yh3) I; daf-2(e1370) III* (unoutcrossed), IJ1591 *daf-18(yh1) IV*, IJ773 *daf-2(e1370) III; daf-18(nr2037) IV* obtained by crossing IJ681 and CF1041, IJ681 *daf-18(nr2037) IV*, IJ1854 *daf-2(e1370) III; daf-18(syb499) IV* obtained by crossing IJ1804 and CF1041, IJ1804 *daf-18(syb499) IV*, IJ1646 *daf-2(e1370) III; daf-18(yh1) IV; yhIs78[daf-18p::mCherry::daf-18 WT; odr-1p::RFP]* obtained by crossing IJ1554 and IJ1592, IJ1554 *daf-2(e1370) III; daf-18(nr2037) IV; yhIs78[daf-18p::mCherry::daf-18 WT; odr-1p::RFP]*, CF2380 *tax-4(p678) III*, IJ385 *daf-2(e1368) III*, IJ1665 *daf-2(e1368) III; daf-18(yh1) IV* obtained by crossing IJ385 and IJ1592, IJ2072 *daf-2(e1368) III; daf-18(nr2037) IV* obtained by crossing IJ385 and IJ773, IJ1926 *daf-2(e979) III*, IJ1349 *daf-2(e979) III; daf-18(yh1) IV* obtained by crossing IJ1926 and IJ1591, IJ1353 *daf-2(e979) III; daf-18(nr2037) IV* obtained by crossing IJ1926 and IJ682, TJ1052 *age-1(hx546) II*, IJ1993 *age-1(hx546) II; daf-18(yh1) IV* obtained by crossing TJ1052 and IJ1591, IJ1089 *age-1(hx546) II; daf-18(nr2037) IV* obtained by crossing IJ265 and IJ682, IJ604 *daf-2(e1370) III; daf-18(e1375) IV* obtained by crossing IJ531 and CF1041, IJ531 *daf-18(e1375) IV*, IJ617 *daf-2(e1370) III; daf-18(ok480) IV* obtained by crossing IJ264 and CF1041, IJ264 *daf-18(ok480) IV*, IJ1855 *daf-2(e1370) III; daf-18(mg198) IV* obtained by crossing SO26 and CF1041, IJ1856 *daf-18(pe407) IV* obtained by crossing JN1483 and CF1041, CF1184 *daf-2(e1370) III; daf-18(mu397) IV*, CF1185 *daf-2(e1370) III; daf-18(mu398) IV*, IJ484 *yhEx94[rpl-28p::CFP::PH$_{AKT}$, odr-1p::RFP]*, yhIs49[rpl-28p::CFP::PH$_{AKT}$, odr-1p::RFP] obtained by UV integration of IJ484, IJ1357 *daf-2(e1370) III; yhIs49[rpl-28p::CFP::PH$_{AKT}$, odr-1p::RFP]* obtained by crossing IJ798 and CF1041, IJ1869 *daf-2(e1370) III; daf-18(yh1) IV; yhIs49[rpl-28p::CFP::PH$_{AKT}$; odr-1p::RFP]* obtained by crossing IJ1357 and IJ1592, IJ1868 *daf-2(e1370) III; daf-18(nr2037) IV; yhIs49[rpl-28p::CFP::PH$_{AKT}$; odr-1p::RFP]* obtained by crossing IJ1357 and IJ773, IJ1058 *daf-16(mu86) I; muIs112[daf-16p::GFP::daf-16cDNA; odr-1p::RFP]*, IJ1456 *daf-16(mu86) I; daf-2(e1370) III; muIs112[daf-16p::GFP::daf-16cDNA; odr-1p::RFP]* obtained by crossing IJ922 and CF2688, IJ1831 *daf-16(mu86) I; daf-2(e1370) III; daf-18(yh1) IV; muIs112[daf-16p::GFP::daf-16cDNA; odr-1p::RFP]* obtained by crossing IJ1671 and IJ1456, IJ1573 *daf-16(mu86) I; daf-2(e1370) III; daf-18(nr2037) IV; muIs112[daf-16p::GFP::daf-16cDNA; odr-1p::RFP]* obtained by crossing IJ921 and IJ1456, IJ1108 *daf-16(mu86) I; daf-2(e1370) III; daf-18(mg198) IV; muIs112[daf-16p::GFP::daf-16cDNA; odr-1p::RFP]* obtained by crossing IJ1573 and IJ1855, IJ1112 *daf-16(mu86) I; daf-2(e1370) III; daf-18(pe407) IV; muIs112[daf-16p::GFP::daf-16cDNA; odr-1p::RFP]* obtained by crossing IJ1573 and IJ1856, IJ1139 *daf-16(mu86) I; daf-2(e1370) III; daf-18(e1375) IV; muIs112[daf-16p::GFP::daf-*

16cDNA; odr-1p::RFP] obtained by crossing IJ1573 and IJ604, IJ1154 *daf-16(mu86) I; daf-2(e1370) III; daf-18(ok480) IV; muIs112[daf-16p::GFP::daf-16cDNA; odr-1p::RFP]* obtained by crossing IJ1573 and IJ617, IJ1157 *daf-16(mu86) I; daf-2(e1370) III; daf-18(mu397) IV; muIs112[daf-16p::GFP::daf-16cDNA; odr-1p::RFP]* obtained by crossing IJ1573 and CF1184, IJ1160 *daf-16(mu86) I; daf-2(e1370) III; daf-18(mu398) IV; muIs112[daf-16p::GFP::daf-16cDNA; odr-1p::RFP]* obtained by crossing IJ1573 and CF1185, IJ1553 *ldIs007[skn-1p::skn-1b/c::GFP; rol-6(su1006gf)]*, IJ1566 *daf-2(e1370) III; ldIs007[skn-1p::skn-1b/c::GFP; rol-6(su1006gf)]* obtained by crossing IJ1553 and IJ785, IJ1816 *daf-2(e1370) III; daf-18(yh1) IV; ldIs007[skn-1p::skn-1b/c::GFP; rol-6(su1006gf)]* obtained by crossing IJ1566 and IJ1592, IJ1570 *daf-2(e1370) III; daf-18(nr2037) IV; ldIs007[skn-1p::skn-1b/c::GFP; rol-6(su1006gf)]* obtained by crossing IJ1553 and IJ785, IJ1013 *daf-2(e1370) III; daf-18(mg198) IV; ldIs007[skn-1p::skn-1b/c::GFP; rol-6(su1006gf)]* obtained by crossing IJ1570 and IJ1855, IJ1046 *daf-2(e1370) III; daf-18(pe407) IV; ldIs007[skn-1p::skn-1b/c::GFP; rol-6(su1006gf)]* obtained by crossing IJ1570 and IJ1856, IJ1070 *daf-2(e1370) III; daf-18(e1375) IV; ldIs007[skn-1p::skn-1b/c::GFP; rol-6(su1006gf)]* obtained by crossing IJ1570 and IJ604, IJ1078 *daf-2(e1370) III; daf-18(ok480) IV; ldIs007[skn-1p::skn-1b/c::GFP; rol-6(su1006gf)]* obtained by crossing IJ1570 and IJ617, IJ1084 *daf-2(e1370) III; daf-18(mu397) IV; ldIs007[skn-1p::skn-1b/c::GFP; rol-6(su1006gf)]* obtained by crossing IJ1570 and CF1184, IJ1086 *daf-2(e1370) III; daf-18(mu398) IV; ldIs007[skn-1p::skn-1b/c::GFP; rol-6(su1006gf)]* obtained by crossing IJ1570 and CF1185, IJ982 *skn-1(lax188) IV*, IJ1981 *daf-2(e1370) III; skn-1(lax188) IV* obtained by crossing IJ982 and IJ1592, IJ1982 *daf-18(yh1) IV skn-1(lax188) IV* obtained by crossing IJ982 and IJ1592, IJ1983 *daf-2(e1370) III; daf-18(yh1) IV skn-1(lax188) IV* obtained by crossing IJ982 and IJ1592, IJ1984 *daf-18(nr2037) IV skn-1(lax188) IV* obtained by crossing IJ982 and IJ773, IJ1985 *daf-2(e1370) III; daf-18(nr2037) IV skn-1(lax188) IV* obtained by crossing IJ982 and IJ773, IJ1625 *skn-1(zj15) IV*, IJ2036 *daf-2(e1370) III; skn-1(zj15) IV* obtained by crossing IJ1625 and IJ1592, IJ2037 *daf-2(e1370) III; daf-18(yh1) IV skn-1(zj15) IV* obtained by crossing IJ1625 and IJ1592, IJ2042 *daf-2(e1370) III; daf-18(nr2037) IV skn-1(zj15) IV* obtained by crossing IJ1625 and IJ773, IJ921 *daf-16(mu86) I; daf-2(e1370) III; daf-18(nr2037) IV* obtained by crossing CF1085 and IJ773, and IJ1671 *daf-16(mu86) I; daf-2(e1370) III; daf-18(yh1) IV* obtained by crossing CF1085 and IJ1592.

**EMS mutagenesis screen**. EMS mutagenesis screen was performed as described previously[60], with modifications. Synchronized L4 larval *daf-2(e1370)* worms were washed with M9 buffer until residual bacteria were cleaned and then exposed to 47 mM ethyl methanesulfonate (EMS, Sigma, St. Louis, MO, USA) in M9 buffer for 4 hrs at 20 °C with rotation. After washing worms three times with M9 buffer, mutagenized P$_0$ worms were placed on OP50-containing chicken egg plates (see below for details) until the majority of F$_1$ worms became adults at 20 °C. The F$_1$ adult worms were then bleached for the synchronization of F$_2$ eggs. With four independent mutagenesis trials, approximately 25,000,000 F$_2$ eggs were transferred onto the OP50-seeded nematode growth medium (NGM) plates and cultured at 25 °C to screen dauer-suppressor mutants. The 269 F$_2$ dauer-suppressor mutants that were recovered as L4 or young adult animals were picked and directly transferred onto plates completely covered with PA14 (big lawn) for screening PA14-resistant worms. Simultaneously, 100 *daf-16(mu86); daf-2(e1370)* mutants were used as a control for each of the four mutagenesis screen trials. When all the *daf-16(mu86); daf-2(e1370)* animals were dead, each of 21 F$_2$ worms that were alive at that point was singled and moved onto an OP50-seeded NGM plate for obtaining F$_3$ animals. Among these 21 singled F$_2$ animals, 18 mutants produced F$_3$ progeny. Among them, 14 animals contained the same allele, *yh1*, and the other four alleles were named as *yh2* through *yh5*. The enhanced resistance against PA14 conferred by *yh4* and *yh5* was not reproduced, and therefore *yh1*, *yh2*, and *yh3* were used for further characterization.

**Preparation of chicken egg plates**. Chicken egg plates were prepared for a large-scale worm culture as follows. Chicken eggs were rinsed with 100% ethanol (DAE-JUNG, Siheung, South Korea), and separated egg yolks were transferred into a sterile beaker. Sterilized double distilled water (25 ml/egg) was mixed with the yolk by using stirrer. To inactivate lysozymes, the egg yolk mixture was incubated at 60 °C for one hr and was subsequently cooled to room temperature. Concentrated *E. coli* OP50 (50X) was mixed with the egg yolk mixture (1:3 ratio) and diluted with M9 buffer. The OP50-yolk mixture was then seeded on 100 mm NGM plates (2 ml/plate).

**Identification of mutated loci**. Complementation tests were performed as described previously[61], with modifications. Specifically, the complementation tests were conducted to determine whether *yh1*, *yh2*, and *yh3* mutations resided in known Daf-d (dauer formation-defective) genes, such as *daf-16* and *daf-18*, whose mutations suppress the dauer formation of *daf-2(e1370)* mutants at 25 °C[5]. Individual L4 hermaphrodite *daf-2(e1370)* animals containing *yh1*, *yh2*, or *yh3* allele were mated with at least eight male *daf-16(mu86); daf-2(e1370)* and *daf-2(e1370); daf-18(nr2037)* mutants. Each complementation test was performed at least twice. The proportion of dauer was calculated by counting the number of dauer worms, non-dauer larvae, and adults among F$_1$ population. After the complementation test was completed, genomic fragments of *daf-18* and *daf-16* were PCR amplified using the genomic DNA of *daf-2(e1370); daf-18(yh1)*, *daf-16(yh2); daf-2(e1370)*, and *daf-16(yh3); daf-2(e1370)* animals obtained from the lysis of worms with proteinase K

(Invitrogen, Carlsbad, CA, USA). The amplified PCR products were cloned into pBluescript II SK(+) (Addgene, Watertown, MA, USA) as described previously[62]. The sequences of these clones were confirmed using Sanger sequencing (Solgent, Daejeon, South Korea).

**Generation of transgenic animals**. *mCherry::daf-18* transgenic animals generated in a previous report[20] was used to rescue the phenotypes of *daf-18(yh1)* animals. *rpl-28p::CFP::PH$_{AKT}$*–expressing animals were generated in this study as follows. For generating *CFP::PH$_{AKT}$* construct, the *CFP::PH$_{AKT}$*-expression vectors (CFP-PH$_{AKT}$)[63,64] were linearized by NheI restriction enzyme and were treated with Klenow (F. Hoffmann–La Roche, Basel, Switzerland) for blunt end generation. The linearized vectors were then digested with XbaI to obtain *CFP::PH$_{AKT}$* DNA fragments. pPD129.57 vectors (L4455, Fire lab *C. elegans* vector kit) containing a *rpl-28* promoter were digested with SmaI and NheI restriction enzymes, and the *CFP::PH$_{AKT}$* fragments were inserted into the linearized pPD129.57 using T4 DNA ligase (New England Biolabs, Ipswich, MA, USA). The transgenic strain was generated by injecting the plasmid (25 ng/μl) and a co-injection marker (*odr-1p::RFP*, 75 ng/μl) into the gonads of day one adults. The extrachromosomal array transgenes were integrated with UV irradiation[65].

**Pathogen resistance assays**. Pathogen resistance assays were performed as described previously[66], with modifications. For small-lawn assays, *Pseudomonas aeruginosa* PA14 was cultured in LB media overnight at 37 °C, and 5 μl of the liquid culture was subsequently seeded on each high-peptone NGM plate (0.35% bacto-peptone). For big-lawn assays, 15 μl of overnight-cultured PA14 was seeded onto each high-peptone NGM plate with a glass spreader. The PA14-seeded plates were cultured at 37 °C for 24 hrs and moved into a 25 °C incubator for 24 hrs before assays. L4-stage worms that were grown on OP50-seeded NGM plates were transferred to PA14 plates containing 50 μM FUDR (5-fluoro-2'-deoxyuridine, Sigma, St. Louis, MO, USA) that prevents progeny from hatching. The assays were performed at 25 °C, and the survival of worms was scored at least once a day. The worms were counted as dead if the worms did not respond to prodding. All the assays were conducted at least twice independently. OASIS (https://sbi.postech.ac.kr/oasis/) and OASIS2 (https://sbi.postech.ac.kr/oasis2/) were used for statistical analysis[67,68], and *p* values were calculated using a log-rank (Mantel–Cox method) test.

**Stress resistance assays**. Stress resistance assays were performed as described previously[20], with modifications. Gravid adults were allowed to lay eggs for 12 hrs on NGM plates seeded with OP50. For the oxidative stress assay, L4-stage worms were transferred onto 5 μM FUDR-treated NGM plates with *E. coli* bacteria and 7.5 mM tert-butyl hydroperoxide (t-BOOH, Sigma, St. Louis, MO, USA) solution. For the thermotolerance assay, L4-stage worms were placed in a 35 °C incubator. The number of live worms was counted every 2 or 3 hr and recorded as dead when the worms did not respond to tactile stimuli with a platinum wire. All the assays were conducted at least twice independently. OASIS (https://sbi.postech.ac.kr/oasis/) and OASIS2 (https://sbi.postech.ac.kr/oasis2/) were used for statistical analysis[67,68], and *p* values were calculated using a log-rank (Mantel–Cox method) test.

**Lifespan assays**. Lifespan assays were performed at 20 °C or 25 °C on NGM plates seeded with OP50 for experiments with mutants or HT115 for experiments using RNAi as described previously[69], with minor modifications. For lifespan assays with FUDR that prevents progeny from hatching, synchronized young (day 1) adult worms were transferred onto 5 μM FUDR-treated NGM plates with *E. coli* and moved onto NGM plates freshly treated with FUDR after 24 hrs. For the experiments without FUDR, young (day 1) adults were placed on new plates every 1–2 days until the worms stopped laying eggs. For RNAi experiments, 1 mM isopropyl-β-D-thiogalactoside (IPTG, Gold Biotechnology, St. Louis, MO, USA) was supplemented onto RNAi bacteria-seeded plates, containing 50–100 μg/ml ampicillin (USB, Santa Clara, CA, USA), and incubated at room temperature for 24 hrs before the assays. The number of live or dead worms was scored every 2 or 3 days until all the animals were dead. Worms that ruptured, displayed internal hatching, burrowed, or crawled off the plates were censored but included for statistical analysis. All the assays were conducted at least twice independently. OASIS (https://sbi.postech.ac.kr/oasis/) and OASIS2 (https://sbi.postech.ac.kr/oasis2/) were used for statistical analysis of lifespan assays[67,68], and *p* values were calculated using a log-rank (Mantel-Cox method) test.

**Dauer formation assays**. Dauer assays were performed as previously described[20], with minor modifications. Gravid adult worms were allowed to lay eggs on NGM plates at 25 °C or 27 °C depending on assay conditions and removed after 3–6 hrs for synchronization of eggs. The F$_1$ progeny were examined for dauer formation after 3 or 4 days at 25 °C or 27 °C. Dauer formation was visually determined[14] under a dissecting stereomicroscope (SMZ645, Nikon, Tokyo, Japan).

**PA14-GFP accumulation assays**. Intestinal accumulation of PA14 expressing GFP (PA14-GFP) was measured as previously described[66], with minor modifications. Big-

lawn PA14 plates, for which the whole surface was covered by the bacteria, were prepared by spreading 15 μl of overnight culture of PA14-GFP in LB media containing 50 μg/ml kanamycin (Sigma, St. Louis, MO, USA) onto NGM plates that contained 0.35% peptone. The NGM plates were incubated at 37 °C for 24 hrs, and subsequently stored at 25 °C for additional 24 hrs before use. L4-stage larvae were infected with PA14-GFP for 36 to 48 hrs. *p* values were calculated by using chi-squared test.

**Microscopy**. Fluorescence images of worms were captured by using Axiocam (Zeiss Corporation, Jena, Germany) mounted on a HRc Zeiss Axioscope A.1 (Zeiss Corporation, Jena, Germany) equipped with EC Plan-Neofluar (Zeiss Corporation, Jena, Germany) objective lens. Green fluorescence was detected by using Zeiss filter set 38 Endow GFP shift free emission filter (Zeiss Corporation, Jena, Germany). Animals used for PA14-GFP accumulation assays or CFP::PH$_{AKT}$ localization experiments were placed on 2% agarose pads and were anesthetized with 100 mM sodium azide (DAEJUNG, Siheung, South Korea) before imaging.

**Measurement of developmental time**. Developmental time was measured as previously described[2], with minor modifications. Adult worms were washed off NGM plates using M9 buffer, and the remaining eggs were incubated at 20 °C for 1–2 hrs. Newly hatched L1-stage worms were transferred onto OP50-seeded plates and were cultured at 20 °C. After 40 hrs of synchronization, the numbers of adult and non-adult worms were counted at 20 °C. Worms that contained at least one egg in their bodies were considered as adults, and the number of adult worms was counted every 2 hrs. The assay was conducted at least twice independently. Two-tailed Student's *t* test was used for statistical analysis.

**Measurement of total brood size**. Total brood size was measured as previously described[2], with minor modifications. A single L4 hermaphrodite was transferred onto an NGM plate seeded with OP50 and maintained at 20 °C. Each of the individual worms was transferred onto a freshly OP50-seeded plate every day until the worms stopped laying eggs for 2 days in a row. The number of viable larvae that reached L4 stage descended from a single hermaphrodite was set as the brood size. The brood size measurement was performed five times independently at 20 °C, and two-tailed Student's *t* test was used for statistical analysis.

**Measurement of swimming**. Swimming rate (body bend in liquid per min) of worms was measured as described previously[9,70], with minor modifications. Ten worms at indicated ages were transferred into a well in 24-well plates containing 1 ml of M9 buffer. After 1 min for stabilization in a new environment, the body bending of worms in liquid was recorded by using a digital microscope (DIMIS-M, Siwon Optical Technology, Anyang, South Korea). The body bends of the individual worms were counted for 30 sec and converted to the number of bending per min. Dead worms were excluded from the assays, and two-tailed Student's *t* test was used for statistical analysis for the measurement of swimming at day 0 (L4-stage worms) and day 7.

**Measurement of moving worms in population**. Percentage of moving worms in population was measured as described previously[12,71], with minor modifications. The movements of age-synchronized worms were categorized as class A, class B, and class C as described previously[71]; class A animals are healthy and mobile with typical sinusoidal movement, class B animals display mobile but irregular movement and require a prodding stimulus for the movement, and class C worms do not move. The movements of worms were scored every 2–3 days starting from L4 stage until all the worms on the plates were dead. Dead worms were excluded from the assays.

**Feeding assays**. Feeding (pharyngeal pumping) rate of worms was measured as described previously[70], with minor modifications. Ten worms at indicated ages grown on OP50 were transferred onto experimental plates containing PA14-GFP or OP50 as indicated. The number of pumping was counted for 30 sec by observing the pharyngeal pumping of a worm under a dissecting microscope, and the measurements were re-scaled to the number of pumping per min. Dead worms were excluded from the assays, and two-tailed Student's *t* test was used for statistical analysis for the measurement of feeding at day 0 (L4-stage worms).

**Quantitative RT-PCR analysis**. Quantitative RT-PCR was performed as described previously[69], with modifications. Synchronized pre-fertile or day 1 adult worms at 20 °C were harvested with M9 buffer, and total RNA was extracted using RNAiso Plus (Takara, Shiga, Japan). cDNA templates were synthesized by using ImProm-II Reverse Transcriptase (Promega, Madison, WI, USA) with random primers. cDNA samples were used for quantitative RT-PCR with SYBR green dye (Applied Biosystems, Foster City, CA, USA) by using StepOne Real-Time PCR System (Applied Biosystems, Foster City, CA, USA). Data were analyzed by using comparative C$_T$ method following the manufacturer's protocol. The average values of *ama-1* or *pmp-3* mRNA levels were used as a control for normalization, and the average of at least two technical repeats was applied for each biological data point. See Supplementary Dataset 5 for primer details.

**RNA sequencing**. Total RNAs were extracted from day 1 adult *daf-2(e1370)*, *daf-2(e1370); daf-18(yh1)*, and *daf-2(e1370); daf-18(nr2037)* worms, four times independently; one dataset using *daf-2(e1370); daf-18(nr2037)* animals was excluded from our analysis due to incorrect genotype. RNA was extracted by using RNAiso Plus (Takara, Shiga, Japan), and was subsequently purified by using 75% ethanol. The qualities of RNA samples were analyzed using 2100 Bioanalyzer (Agilent, CA, USA). The RNA integrity numbers, which indicate the quality of RNA samples, of all the samples were sufficiently high (> 8.8) for RNA sequencing. cDNA libraries were generated by using reverse transcription of RNA samples and paired-end sequencing of Illumina platform was performed (Macrogen, Seoul, South Korea).

**Analysis of RNA-sequencing data**. mRNA-sequencing analysis was performed as described previously[72], with modifications. Sequencing pairs were aligned to the *C. elegans* genome WBcel235 (ce11) and Ensembl transcriptome (release 95) by using STAR (v.2.7.0e). Aligned pairs on genes were quantified by using RSEM (v.1.3.1). Alignment and quantification of RNA-seq data in this study were conducted based on the parameters described in the guidelines of ENCODE long RNA-Seq processing pipeline (https://www.encodeproject.org/pipelines/ENCPL002LPE/). The batch effects of samples were removed by upper-quartile normalization followed by RUVSeq (v.1.16.1) with internal control genes. Global expression changes of previously published gene sets in a comparison *daf-2(e1370)* vs. *daf-2(e1370); daf-18(nr2037)* or *daf-2(e1370)* vs. *daf-2(e1370); daf-18(yh1)* were represented as normalized enrichment scores (NES) by using gene set enrichment analysis (GSEA) (v.3.0) or calculating cumulative fractions with read counts of all expressed genes. Gene sets whose false discovery rate $q$ value < 0.05 in any comparison were regarded as significant in GSEA. The significance of difference in calculating cumulative fractions was computed by using two-tailed paired permutation test using asymptotic approximation. Differentially expressed genes (fold change > 2 and adjusted $p$ value < 0.05) were identified by using DESeq2 (v.1.22.2). Wald test $p$ values were adjusted for multiple testing using the procedure of Benjamini and Hochberg. Gene ontology terms enriched in genes whose expression changes were greater by *daf-18(−)* (fold change > 2) than by *daf-18(yh1)* in *daf-2(e1370)* mutants were identified by using GOstats (v.2.48.0), and summarized by using Revigo. Subsequently, these genes were compared to the genes expressed in different tissues based on Worm tissue[73]. R (v.3.6.1, http://www.r-project.org) was used for plotting data.

**Preparation of recombinant PTEN protein**. Recombinant PTEN expression was performed as described previously[74]. Wild-type and mutant *PTEN* constructs (WT, C105Y, and C124S) were subcloned into pFastBac containing an N-terminal six histidine tag. pFastBac-HTA was digested with EcoRI and HindIII. Using pCMV-FLAG-PTEN (Addgene, Watertown, MA, USA) as a template, a 1.2 kb fragment containing human *PTEN* was amplified with primers including digestion sites (forward 5′-GCGCCATGGATCCGGAATTCATGACAGCCATCATCAAAGA-3′ and reverse 5′-GTACTTCTCGACAAGCTTTCAGACTTTTGTAATTTGTG-3′). The PCR product was then subcloned into pFastBac-HTA. Recombinant PTEN proteins were expressed in Sf9 cells with the Bac-to-Bac expression system (Invitrogen). Sf9 cells were transfected with recombinant bacmid for 72 hrs, and the cell culture media containing baculoviruses were then harvested. Following the baculovirus infection, the cell pellets were resuspended in buffer A [20 mM Tris-HCl (pH 7.5), 50 mM NaCl, and 1 mM DTT], and underwent the Ni-NTA affinity chromatography for purifying his-tagged PTEN proteins. Protein fractions were further dialyzed in buffer B [25 mM Tris-HCl (pH 7.5), 100 mM NaCl, and 1 mM DTT]. Purified proteins were then separated on a 10% SDS-PAGE gel with Coomassie blue staining, and the proteins that displayed appropriate sizes were confirmed by using western blotting with an anti-His antibody (1:1,000, #2365, Cell signaling technology, Danvers, MA, USA).

**In vitro phosphatase assays**. In vitro lipid and protein phosphatase assays were performed with a malachite green phosphatase assay kit (K-1500, Echelon Biosciences, Salt Lake City, UT, USA) following the manufacturer's instruction. For lipid phosphatase assays, soluble phosphatidylinositol 3,4,5-trisphosphate diC8 (PIP$_3$ diC8, Echelon Biosciences, Salt Lake City, UT, USA) was diluted to 1 mM in distilled water. Indicated amounts of purified PTEN recombinant proteins were incubated with 3 μl of the 1 mM PIP$_3$ solution in 25 μl Tris-buffer [25 mM Tris-HCl (pH 7.4), 140 mM NaCl, 2.7 mM KCl and 10 mM DTT] for 40 min at 37 °C. For protein tyrosine phosphatase assays, a synthetic phospho-tyrosine peptide (YEEEEpYEEEE) was used as a substrate[75]. Indicated amounts of recombinant His-PTEN proteins (WT, C105Y, and C124S) and the protein tyrosine phosphatase 1B (PTP1B; positive control, R&D system, Minneapolis, MN, USA) were used as enzymes. The peptide substrates (100 μM) were incubated with each enzyme in 25 μl reaction buffer [50 mM Tris-HCl (pH 7.4), 10 mM DTT] for 60 min at 37 °C. One hundred microliters of malachite green solution was then added to terminate the enzyme reaction and incubated for 20 min at room temperature. The released phosphate was quantified by measuring absorption spectrum at 620 nm using a microplate reader. A standard curve derived from 0.1 mM phosphate provided by the assay kit was used to convert the absorbance value at 620 nm to amount of free phosphate. To obtain dose-response curves of PTEN enzymes, the protein concentration was increased stepwise by 2-fold to reach a final 10 nM concentration for the protein phosphatase assay. The dose-dependent activity was not observed for the phosphatase activity-dead variant, PTEN$^{C124S}$.

**Measurement of subcellular localization of CFP::PH$_{AKT}$, DAF-16::GFP, and SKN-1::GFP**. Synchronized L4-stage worms with CFP::PH$_{AKT}$ localized in the membrane were counted, and percentage of worms with membrane-localized CFP::PH$_{AKT}$ were calculated. $p$ values were calculated using two-tailed Student's $t$ test by comparing mean values of experimental group with that of control group. Subcellular localization of fluorescence fusion proteins was determined using fixed worms as described previously[69]. Briefly, stage-synchronized worms were harvested with M9 containing 0.01% polyethylene glycol 4000 (PEG 4000, Tokyo Chemical Industry, Tokyo, Japan) and were washed three times. The animals were fixed with 4% paraformaldehyde (158127, Sigma, St. Louis, MO, USA) solution in phosphate-buffered saline (PBS, AM6924, Thermo Fisher Scientific, MA, USA) [137 mM NaCl, 2.7 mM KCl, 8 mM Na$_2$HPO$_4$, and 2 mM KH$_2$PO$_4$] for 45 min with gentle agitation. Worms were washed with PBS twice for five min with gentle agitation for each washing step. Fixed worms were then stored in 70% ethanol overnight at 4 °C. Worms were placed on a 2% agarose pad on a slide glass, and the agar pad was covered with a coverslip before imaging. L2- or L3-stage worms and L4-stage worms were used for determining subcellular localization of DAF-16::GFP and SKN-1::GFP, respectively. Quantification of subcellular localization of DAF-16::GFP and SKN-1::GFP were conducted as previously described[76,77], with some modifications. Subcellular DAF-16::GFP localization was scored as follows. Nuclear: predominant nuclear localization, intermediate: partial nuclear localization, cytosolic: predominant cytosolic localization. For the experiments measuring the subcellular localization of SKN-1::GFP, worms with SKN-1::GFP in the nuclei of intestinal cells at high, medium, and low levels were determined as follows. High: > 50% of the nuclei with SKN-1::GFP, medium: < 50% of the nuclei with SKN-1::GFP, low: very dim GFP in the nuclei. $p$ values were calculated using chi-squared test by comparing each mean value of experimental group with that of the control group. All subcellular localization assays were performed double-blindly by at least two independent researchers.

**Western blot assays**. Western blot assay was performed as described previously with minor modifications[78]. Briefly, bleached eggs were placed on *E. coli* OP50-seeded NGM plates. Hatched worms from the eggs were then allowed to develop to pre-fertile or day 1 young adults at 20 °C, and were subsequently washed three times by using M9 buffer. The worms were then frozen in liquid nitrogen with 2X Laemmli sample buffer (#161-0747, Bio-Rad, Contra Costa County, CA, USA) containing 5% 2-mercaptoethanol (M3148, Sigma, St. Louis, MO, USA), boiled for 10 min at 98 °C, and vortexed for 10 min. After centrifugation of the worm lysates at 15,871 $g$ for 30 min, the supernatants were loaded to 10% SDS-PAGE. The proteins were then transferred to PVDF membrane (#10600021, GE healthcare, Chicago, IL, USA) at 300 mA for 1 hr. The membranes were incubated with 5% bovine serum albumin solution in 1x TBS-T [24.7 mM Tris-HCl (pH 7.6), 137 mM NaCl, 2.7 mM KCl and 0.1% Tween 20] for blocking at room temperature for 30 min. Primary antibodies against phospho-PMK-1 (1:1,000, #9211, Cell Signaling Technology, Danvers, MA, USA), α-tubulin (1:2,000, sc-32293, Santa Cruz Biotechnology, Dallas, TX, USA), or His (1:1,000, #2365, Cell signaling technology, Danvers, MA, USA) were incubated with the membranes overnight at 4 °C with gentle agitation. The membranes were then washed four times for 15 min using 1x TBS-T followed by incubating with secondary antibodies against rabbit (1:5,000, #SA8002, ABfrontier, Seoul, South Korea) or mouse (1:5,000, #SA8001, ABfrontier, Seoul, South Korea). After washing membranes with 1x TBS-T for 15 min, the membranes were then treated with ECL substrate (#1705061, Bio-Rad, Contra Costa County, CA, USA) for detecting protein bands. Images were visualized with ChemiDoc XRS$^+$ system (Bio-Rad, Contra Costa County, CA, USA), and analyzed by using Image Lab software (Bio-Rad, Contra Costa County, CA, USA).

**Reporting summary**. Further information on research design is available in the Nature Research Reporting Summary linked to this article.

## Data availability

RNA-seq raw data and processed data are available at Gene Expression Omnibus under accession code GSE154338. The *C. elegans* databases used in this study were WormBase release (WS233) and UniProt release (2020_01). Transcriptomics data are summarized in Supplementary Dataset 1. Survival data are placed in Supplementary Dataset 2. Dauer and health span measurement data are collected in Supplementary Dataset 3 and 4, respectively. Primers used for quantitative RT-PCR in this study are included in Supplementary Dataset 5. All relevant data, strains, or plasmids are available from the corresponding author on reasonable request. Source data are provided with this paper.

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

## Acknowledgements

The authors thank Drs. Cynthia Kenyon, Keith Blackwell, Gary Ruvkun, Yuichi Iino, and Yongsoon Kim, and the *Caenorhabditis* Genetics Center (CGC) for providing some *C. elegans* strains, and Dr. Sean P. Curran for providing access to RNA-seq data using *skn-1(lax188)* animals. We also thank Dr. Ji-Joon Song, Gijeong Kim, and Yoojoong Kim for their help and advice for baculovirus expression system. We thank all the Lee lab members for comments on the manuscript and discussion. This work was supported by the Korean Government (MSICT) through the National Research Foundation of Korea (NRF) NRF-2019R1A3B2067745 to S-J.V.L. and NRF-2020R1C1C1013546 to E.K.

## Author contributions

H.-E.H.P., W.H., S.H., E.K. and S.-J.V.L. designed the experiments; H.-E.H.P., W.H., S.H., E.K., O.A., S.P., H.G.S., Y.L., D.L. and S.-J.V.L. performed the experiments; H.-E.H.P., W.H., S.H. and S.-J.V.L. analyzed the data; H.-E.H.P., W.H., S.H., E.K. and S.-J.V.L. wrote the manuscript; W.D.H. provided critical reagents for key experiments; and S.-J.V.L. contributed to the supervision.

## Competing interests

The authors declare no competing interests.
