## [Peer Review File · Nature Communications]

REVIEWER COMMENTS

Reviewer #1 (Remarks to the Author):

This is a very interesting story. Reducing IIS signalling is one of the most effective mechanisms to extend lifespan and stress resistance from invertebrates to mammals. However, reduced IIS also induces pleiotropic effects such as impaired motility, development and reproduction in *C. elegans*. Thus, defining mechanisms that can maintain the beneficial effects of reduced IIS but eliminate its detrimental effects are of central importance in aging research. Here the authors perform a series of elegant and solid experiments that indicate that a single amino acid change in DAF-18/PTEN (*daf-18(yh1)*) enables to maintain longevity and stress resistance without defects in growth and motility. They further demonstrate that *daf-18(yh1)* retained the activity of DAF-16 but diminished the detrimental upregulation of SKN-1. I only have a few comments that I hope the authors will be able to address:

- The central part of the abstract should be re-written to make it more lineal and organized.
- In Fig. 1E, the authors could do the lifespan lines thinner to make more visibly the WT condition in the graph.
- Besides *daf-2(e1370)* mutant worms, the authors should confirm in other strains expressing distinct mutant alleles of *daf-2* or treated with *daf-2* RNAi whether *daf-18(yh1)* also has mild effects on longevity as well as pathogen, heat stress, and oxidative resistance phenotypes. In other words, if other double *daf-18(yh1)*, *daf-2* mutant (or *daf-2* RNAi) are still long-lived and more resistance to PA14 when compared with WT worms. This is necessary to discard that the effects of *daf-18(yh1)* are specific for *daf-2(e1370)* allele.
- The link between *daf-18(yh1)* and SKN-1 is very interesting and supported by the data provided by the authors. However, it is surprising that *daf-18(yh1);daf-2* mutants still exhibit significant localization of DAF-16 in the nucleus, whereas the effects of *daf-18(yh1)* on the expression levels of DAF-16-regulated genes in *daf-2* mutants are similar to those of *daf-18(-)*. The authors should further discuss these results in the Discussion section.
- I recommend that the authors add a scheme at the end of the manuscript to make more clear their findings for potential readers which are not experts on lifespan regulation by DAF-16/SKN-1

David Vilchez

Reviewer #2 (Remarks to the Author):

Loss of function mutations in the *C. elegans* insulin receptor (DAF-2) increases longevity and resistance to stressors such as oxidative stress and pathogens, However with this increased longevity and stress resistance, there are pleiotropic effects that cause a reduction in health span eg. low brood sizes, low feeding and motility. This manuscript aims to provide data to uncouple the association between longevity and reduced fitness seen in the reduction on of insulin signaling. Here Park et al. report a novel hypomorphic *daf-18/pten* allele, *daf-18(yh1)*, that suppresses the healthspan defects in a hypomorphic *daf-2/insulin receptor* mutant *daf-2(e1370)*, but maintains its long lifespan and resistance to stressors such as oxidative stress and pathogenic bacteria (i.e. *Pseudomonas aeruginosa*). The *daf-18(yh1)* allele was identified in an EMS mutagenesis screen for suppressors of the *daf-*

2(e1370) temperature sensitive dauer constitutive phenotype. It is important to note that the daf-2(e1370) insulin receptor mutant is not a null allele and does retain some activity.

The daf-18(yh1) behaves differently from the null allele daf-18(nr2037) in that the daf-18(-) null in addition to suppressing the dauer constitutive phenotype, also suppresses the adult longevity, pathogen resistance.

Transcriptional analysis shows that the daf-18(yh1) behaves differently than the daf-18 null in the daf-2(-) background, and in particular the daf-18(yh1) seems to have only a minor effect on the overactive DAF-16/FOXO which might explain why longevity is still maintained in the daf-2(-); daf-18(yh1) double. In this study, the author show that daf-18(null) increases SKN-1/NRF activity in *C. elegans* with reduced IIS (daf-2(e1370)), which contributes to shortened lifespan and health span, in contrast the daf-2(-); daf-18(yh1) did not show this increased SKN-1/NRF activity and explains why longevity is retained and health span defects suppressed.

The daf-18(yh1) allele encodes a missense mutation in a conserved cysteine residue in the phosphatase domain (C150Y). The evidence provided suggests that daf-18(yh1) is a hypomorphic mutation, that is to say it retains some activity and not a null allele. Since DAF-18 like its human counterpart, PTEN, antagonizes IIS at the level of PI3K/AGE-1 then loss of daf-18 function leads to increased IIS. As I read the manuscript I could not help think that that the differences in phenotypes of daf-2(-); daf-18(yh1) versus daf-2(2(-); daf-18(null) could simply be explained by the amount of IIS signaling in the daf-2(e1370) is just greater in the daf-18 null when compared to the daf-18(yh1). To the authors credit, they do bring up the hypothesis that maybe discrete thresholds exist for the suppression of different IIS-regulated phenotypes by a weaker allele, daf-18(yh1), compared with a stronger allele, for example daf-18(-). daf-2(-) mutants may exhibit a lower threshold for the suppression of dauer formation than the enhanced pathogen resistance and longevity. To address this, they proposed that multiple hypomorphic mutations should also behave like daf-18(yh1) ie suppress dauer but not the PA14 resistance. They tested 6 alleles of daf-18 and all of them completely suppressed both dauer formation and PA14 resistance (i.e. behaved like the null). One exception was a missense mutant allele, daf-18(pe407) that suppressed the PA14 resistance but not the dauer constitutive daf-2(e1370) phenotype. It was not stated what the missense mutation is in the pe407 allele. This should be included.

The authors conclude that because all 6 daf-18 alleles, with the one exception, suppressed both phenotypes that it refutes the possibility that the threshold for the suppression of constitutive dauer phenotype in daf-2(-) mutants is lower than that of enhanced pathogen resistance and conclude the yh1 allele appears to be a specific reduction-of-function daf-18 allele for retaining immunity and longevity while suppressing dauer formation in daf-2(-) mutants.

I don't think they have ruled out the IIS threshold level hypothesis, and in fact the daf-2(e1370) allele itself shows temperature sensitivity for the dauer constitutive phenotype indicating that the IIS levels may be different at different temperatures supporting differences in threshold levels of IIS-regulated phenotypes. Since we do not know what is the functional consequences of the 6 daf-18 alleles tested are, it is impossible to conclude that there are no IIS threshold levels for different IIS related processes. All 6 alleles tested could reduce DAF-18 function enough such that threshold levels of IIS are allowed to suppress the dauer, and PA resistant phenotypes.

How do these 6 daf-18 alleles behave with respect to the daf-2(e1370); DAF-16::GFP localization? Do they affect the nuclear localization of DAF-16::GFP in daf-2(-) mutants at an intermediate level between daf-18(+) and daf-18(-) (Fig. 4b,c). If the 6 alleles behave like daf-18 null for DAF-16::GFP localization that could explain why PA14 resistance is suppressed with these daf-18 alleles. The DAF-16 and SKN-1 localization should be included for these alleles or some more phenotypic characterization order them in an allelic series.

Genetics with skn-1gf alleles and pmk-1 (upstream activator of SKN-1) suggested that

hyperactivation of SKN-1/NRF contributes to the short lifespan of *daf-2(-); daf-18(-)* worms. In *daf-2(-); daf-18(yh1)* the hyperactivation of SKN-1 is not seen explaining why *daf-18(yh1)* (in addition to DAF-16 activation) does not suppress the longevity phenotype in the *daf-2(-)* mutants. If this was the case why not test *skn-1* loss of function mutants or RNAi on *daf-2(-); daf-18(-)* animals to see if this increases the longevity?

What are the phenotypes of the *daf-18(yh1)* alone? It is mentioned that alone they have reduced healthspan, but the authors should show the data on the *daf-18(yh1)* single mutant compared to the *daf-18(null)* on longevity and, dauer formation. How does *daf-18(yh1)* alone compare to a *daf-18 null* for these classic IIS phenotypes is important information and will support the hypomorphic nature of the *yh1* allele.

Only one *daf-2* allele was used in this study (e1370). Does *daf-18(yh1)* suppress the *daf-2 null* (ie *daf-2(e979)*) for longevity? What about other *daf-2* alleles eg. The alleles that affect the receptor binding domain?

DAF-18/PTEN antagonizes the the PI3k/AGE-1 PI kinase does *daf-18(yh1)* behave differently than *daf-18(null)* with respect to suppression of *age-1* loss of function phenotypes or is the the suppressors phenotypes unique to *daf-2*?

The novel finding in this manuscript is that the authors have identified a point mutation in DAF-18/PTEN that uncouples longevity from impaired fitness in *C. elegans* with reduced IIS.

The *daf-18(yh1)* changes a conserved cysteine to a tyrosine in the phosphatase domain (C150Y). The authors propose that the *yh1* is a specific reduction of function *daf-18* allele that allows the DAF-18 protein to retain pathogen immunity and longevity, while suppressing dauer formation in *daf-2(-)* mutants.

However this could still be argued that the *yh1* is a hypomorph and that it retains some function and thus still has the ability to reduce the IIS that is active in the *daf-2(e1370)*, thereby changing the IIS levels in the *daf-2 (e1370)* mutant. This is consistent with the threshold level of IIS regulating different processes.

Where this manuscript falls short is that if the C150Y is really a special mutation, then how does this mutation give DAF-18/PTEN its special properties? Figure 3a-c show that the C150Y does indeed affect DAF-18/PTEN lipid phosphatase activity. But when you compare the *yh1* to the *daf-18 null* they both reduce the lipid phosphatase activity to similar levels and not significantly different. The analogous human PTEN C105Y version was created but it also shows reduction in lipid phosphatase activity to the well studied PTEN C124S phosphatase-dead variant. So the C150Y behaves similar to a null for lipid phosphatase activity. This questions whether DAF-18 C150Y has lipid phosphatase independent activity. Note that feeding and brood size suppression is not significantly different than the *daf-18 null* and suggests that *yh1* reduces DAF-18 function enough to allow *daf-2(e1370)* IIS to suppress these phenotypes.

Since PTEN has also been shown to have protein phosphatase and important question to address is whether the DAF-18 C150Y or PTEN C105Y retains this protein phosphatase activity? This would make the case that *yh1* is a special loss of function.

The last paragraph has some bold statements such as suggesting “the introduction of a specific *yh1*-like mutation in PTEN to mammalian mutants defective in IIS may boost fitness.” This section should be toned down, as the analogous PTEN Y105C on its own in humans has been associated with sporadic Bannayan-Riley-Ruvalcaba Syndrome.

Reviewer #3 (Remarks to the Author):

In this article, Park et al., describe an allelic variant of the *C. elegans* gene *daf-18* that uncouples improved longevity and immunity from developmental defects observed in mutants of the insulin/IGF1 receptor, *DAF-2*. *daf-18* encodes the worm homolog of the tumor suppressor PTEN phosphatase. Unlike strong *daf-18* loss of function/null mutants, that abolishes *daf-2* longevity and immunity as well as developmental defects, the authors identified a new *daf-18* allele, *yh1*, that minimally affects longevity and immunity but protects the mutants from many of the other defects in developmental fitness measures. The allele is mapped to the phosphatase domain of the protein and the authors' data suggest that it reduces phosphatase activity. Based on gene expression profiling and molecular genetic studies, the authors suggest that *yh1* produces the benefits it does by impacting the activities of two of the transcription factors employed in *daf-2* mutants longevity- by 'maintaining' the activity of the FOXO protein *DAF-16* and 'preventing harmful activation' of the NRF2 homolog, *SKN-1*. The study focuses on a highly topical and important area, the molecular distinctions between lifespan and individual aspects of health and fitness, that is likely to be of great interest to readers in the aging field.

The work is based on highly interesting observations, the experiments are well designed, logical and conducted with rigor. One concern is that while the data is compelling in documenting the selective impact of the *daf-18(yh1)* mutation on longevity/immunity vs. developmental phenotypes, there is limited mechanistic insight into how and why this occurs (especially #1, 2 below) and the evidence supporting some key conclusions is rather weak.

1. How the *yh1* mutation different/impacting *DAF-18* function as compared to the null/other alleles? The data with the PH-domain reporter and human PTEN is very encouraging in implicating altered phosphatase activity. But, both the null and *yh1* show similar effects in this assay. Other mutations that completely suppress dauer formation and immunity also map to the phosphatase domain (*ok480*) as does an allele that selectively/largely suppresses immunity alone (*pe407*). Since the *yh1* and CRISPR alleles do show modest reduction in immunity, and in the absence of data comparing/characterizing the impacts of the different alleles on phosphatase activity, the possibility that *yh1* is simply a weaker allele that crosses the threshold for dauer formation but not immunity cannot be overruled.

2. The evidence supporting the authors' conclusion that *yh1* 'retains *DAF-16* activity but restricts detrimental *SKN-1* upregulation' is weak in premise and experimental strength and described in a very confusing manner. The comparisons of *DAF-16*-, *SKN-1*- mediated transcriptomes with those governed by *daf-9(yh1)* and *daf-9(-)* appear to suggest that the two *daf-9* alleles have the most differential impact on genes both positively and negatively regulated by *PMK-1* signaling followed by *SKN-1*; effects on *DAF-16* targets seem to be the same. *PMK-1* is critical for *daf-2* mutants' immune resistance and directs multiple downstream transcription factors besides *SKN-1* and *DAF-16*. It is very surprising then that *daf18* null appears to cause up-regulation of *PMK-1* targets in *daf-2*. Wouldn't one expect it to suppress them? And if this is indication of toxic 'hyperactivation' then, the data would warrant greater exploration of *PMK-1* modulation in this process, not just *SKN-1*. How does *pmk-1* null/RNAi impact the immunity of *daf-2;yh1*? What does its phosphorylation status look like in these animals vs. the double mutant with the null? Lastly, the 'differential' effects of *daf-16* RNAi and *skn-1(gof)* on survival on pathogen of *daf2;yh1* vs. *daf-2>null* (Fig. 5) cannot be used to draw the strong conclusions arrived at here. *daf-16* RNAi suppressing *daf-2;yh1* mutants long lifespan is taken as evidence that *DAF-16* activity is maintained in *daf-2* mutants. what about *skn-1*

rnai/mutation?

3. The 'transcriptomic analyses' comparing daf-2 gene expression changes governed by different factors (Figs 4 and S6) are described with little clarity in the results section and the phrasing is very confusing about genes positively or negatively regulated by a given factor. It is not explicit that this was harnessing data generated by previous studies. More importantly, the rationale for choosing the many genes is not elaborated. The presentation of the data in the figures can be improved as well. The most striking observation is the differential impact of the two daf-9 alleles on 'up' and 'down' PMK-1 targets and then SKN-1 target. This can be shown in the main figures while the data with the remaining can be moved to the supplement.

4. In the manuscript (eg., lines 25, 64, 35) the authors use multiple assays as 'fitness parameters'- development time, dauer entry, brood size, stress response, mobility etc.,. While technically this may be correct, it makes for confusing reading, especially since healthspan/fitness measures in the aging field usually refer to aspects of adult physiology. It may be helpful to distinguish developmental fitness parameters (development, dauer) from adult functionality metrics (mobility, immunity), especially in describing the impact of the yh1 on them.

5. The alleles yh2 and yh3 that map to daf-16 appear to completely suppress daf-2 immunity unlike yh1 (Fig. 1b-d) and in any case are not investigated further in the article. This data can be removed/moved to the supplement and replaced with data showing the DAF-18::mCherry construct dauer/immunity rescue data (Fig. S2).

6. Data on the syb499 mutants survival on PA14 is not shown in Fig. 1 as mentioned (line 107).

7. Many of the supplementary tables (eg. S1-4) can be converted to .pdf versions/incorporated into the supplementary data file to make them easily accessible to the readers.

8. The article requires some careful editing for nomenclature (eg. line 2: daf2/insulin/IGF1; lines 2, 22, 23: the mammalian homologs are FOXO3A and NRF2) and phrasing/sentence construction (eg., line 40: 'survival against ...stresses'; line 46: delete 'mutant'; line 104)

Reviewer #1 (Remarks to the Author):

This is a very interesting story. Reducing IIS signalling is one of the most effective mechanisms to extend lifespan and stress resistance from invertebrates to mammals. However, reduced IIS also induces pleiotropic effects such as impaired motility, development and reproduction in *C. elegans*. Thus, defining mechanisms that can maintain the beneficial effects of reduced IIS but eliminate its detrimental effects are of central importance in aging research. Here the authors perform a series of elegant and solid experiments that indicate that a single amino acid change in DAF-18/PTEN (*daf-18(yh1)*) enables to maintain longevity and stress resistance without defects in growth and motility. They further demonstrate that *daf-18(yh1)* retained the activity of DAF-16 but diminished the detrimental upregulation of SKN-1. I only have a few comments that I hope the authors will be able to address:

- The central part of the abstract should be re-written to make it more lineal and organized.

> We thank the reviewer's comment. We revised the abstract for enhancing legibility.

Abstract, page 2, line 19: "Insulin/IGF-1 signaling (IIS) regulates various physiological aspects in numerous species. In *Caenorhabditis elegans*, mutations in the *daf-2*/insulin/IGF-1 receptor dramatically increase lifespan and immunity, but generally impair motility, growth, and reproduction. Whether these pleiotropic effects can be dissociated at a specific step in insulin/IGF-1 signaling pathway remains unknown. Through performing a mutagenesis screen, we identified a missense

mutation *daf-18(yh1)* that alters a cysteine to tyrosine in DAF-18/PTEN phosphatase, which maintained the long lifespan and enhanced immunity, while improving the reduced motility in adult *daf-2* mutants. We showed that the *daf-18(yh1)* mutation decreased the lipid phosphatase activity of DAF-18/PTEN, while retaining a partial protein tyrosine phosphatase activity. We found that *daf-18(yh1)* maintained the partial activity of DAF-16/FOXO but restricted the detrimental upregulation of SKN-1/NRF2, contributing to beneficial physiological traits in *daf-2* mutants. Our work provides important insights into how one evolutionarily conserved component, PTEN, can coordinate animal health and longevity.”

- In Fig. 1E, the authors could do the lifespan lines thinner to make more visibly the WT condition in the graph.

> We thank the reviewer’s comment. We made all the survival curve lines thinner for improving visualization.

- Besides *daf-2(e1370)* mutant worms, the authors should confirm in other strains expressing distinct mutant alleles of *daf-2* or treated with *daf-2* RNAi whether *daf-18(yh1)* also has mild effects on longevity as well as pathogen, heat stress, and oxidative resistance phenotypes. In other words, if other double *daf-18(yh1)*, *daf-2* mutant (or *daf-2* RNAi) are still long-lived and more resistance to PA14 when compared with WT worms. This is necessary to discard that the effects of *daf-18(yh1)* are specific for *daf-2(e1370)* allele.

> We appreciate the reviewer's comment. To address whether *daf-18(yh1)* also retained longevity and enhanced pathogen resistance caused by *daf-2* mutant alleles other than *daf-2(e1370)*, we performed lifespan and PA14 resistance assays using animals containing a class 1 *daf-2(e1368)* mutant allele, a class 2 *daf-2(e979)* allele, and *daf-2* RNAi (Supplementary Fig. 3a-f) (Gems et al., 1998; Patel et al., 2008). We found that *daf-18(yh1)* partially maintained longevity in *daf-2(RNAi)* and *daf-2(e979)* worms (Supplementary Fig. 3a,e). These results suggest that the effects of *daf-18(yh1)* on lifespan and immunity in animals with reduced insulin/IGF-1 signaling are not specific to *e1370*. In contrast, we found that *daf-18(yh1)* did not retain the longevity and enhanced pathogen resistance in *daf-2(e1368)* mutants, the class 1 *daf-2* mutants that display weaker longevity and stress resistance phenotypes than *daf-2(e1370)* or *daf-2(RNAi)* (Supplementary Fig. 3c,d) (Gems et al., 1998; Patel et al., 2008; Ewald et al., 2015). One possible explanation for these results is that a certain magnitude of lifespan-extending effect by *daf-2* RNAi or mutation is required for the maintenance of longevity or enhanced immunity by *daf-18(yh1)* in animals reduced with insulin/IGF-1 signaling. In any case, these results indicate that *daf-18(yh1)* can retain lifespan and pathogen resistance in worms with reduced insulin/IGF-1 signaling in multiple conditions and that the effect of *daf-18(yh1)* is not specific for *daf-2(e1370)*. We added these new findings to the manuscript as follows,

Results, page 7, line 130: "We then tested the effects of *daf-18(yh1)* on lifespan and PA14 resistance conferred by other mutations in *daf-2*, a weak *daf-2(e1368)* allele and a strong ligand-binding domain-defective *daf-2(e979)* allele, by RNAi knockdown of *daf-2*^{21,22}, and by a hypomorphic *hx546* allele of *age-1*, which

encodes phosphoinositide 3-kinase that counteracts DAF-18/PTEN phosphatase^{5,6}. We found that *daf-18(yh1)* partially maintained longevity and enhanced immunity in *daf-2(RNAi)* and *daf-2(e979)* worms (Supplementary Fig. 3a,b,e,f), but did not in *daf-2(e1368)* or *age-1(hx546)* mutants (Supplementary Fig. 3c,d,g,h). These results suggest that *daf-18(yh1)* can maintain lifespan and pathogen resistance in animals with reduced IIS caused by genetically inhibited *daf-2* with multiple intervention modes, not specific to *daf-2(e1370)*. In addition, because *daf-2(e1368)* and *age-1(hx546)* mutations cause weak longevity and stress resistance phenotypes (Supplementary Fig. 3c,d,g-j)²¹⁻²⁵, we propose that a certain level of IIS reduction is required for *daf-18(yh1)* to maintain longevity and immunity.”

Supplementary Figure 3 legends, page 6, line 55: “**Supplementary Figure 3: *daf-18(yh1)* retains lifespan and pathogen resistance in worms with reduced IIS in multiple conditions.** (a-h) The effects of *daf-18(yh1)* and *daf-18(nr2037)* [*daf-18(-)*] on lifespan at 20°C (n ≥ 270 for each condition; n = 156 for *daf-2(e979)* animals) and pathogen resistance (n = 180 for each condition; n = 35 for *daf-2(e979)* animals) of *daf-2(RNAi)* (a,b), *daf-2(e1368)* (c,d), *daf-2(e979)* (e,f), and *age-1(hx546)* [*age-1(-)*] (g,h) animals. (i,j) Swimming rate (n = 30 for each condition, body bends per minute in liquid measured from three independent trials) of day 0 (i) and day 7 adult (j) wild-type (WT), *age-1(-); daf-18(+)*, *age-1(-); daf-18(yh1)*, and *age-1(-); daf-18(-)* animals. Error bars indicate the standard error of mean (s.e.m., **p* < 0.05, ***p* < 0.01, n.s.: not significant, two-tailed Student’s *t*-test relative to WT unless otherwise noted).

See Supplementary Tables 1 and 4 for additional repeats and statistical analysis for the survival assay and health span assay data shown in this figure.”

- The link between *daf-18(yh1)* and SKN-1 is very interesting and supported by the data provided by the authors. However, it is surprising that *daf-18(yh1);daf-2* mutants still exhibit significant localization of DAF-16 in the nucleus, whereas the effects of *daf-18(yh1)* on the expression levels of DAF-16-regulated genes in *daf-2* mutants are similar to those of *daf-18(-)*. The authors should further discuss these results in the Discussion section.

> We appreciate the reviewer for raising this issue. We discussed this issue in the Discussion section as follows,

Discussion, page 17, line 343: “Our data indicate that *daf-18(yh1)* retains nuclear localization of DAF-16/FOXO in *daf-2(-)* mutants (Fig. 5f,g), while downregulating DAF-16/FOXO-induced genes in *daf-2(-)* mutants similarly to *daf-18(-)* (Fig. 5a). Thus, the nuclear localization status of DAF-16 does not appear to always correlate with target gene expression levels. One possible interpretation for this is that target gene expression of DAF-16/FOXO is affected by the surrounding density of cofactors or inhibitors of the DAF-16/FOXO. In addition, DAF-16-binding elements (DBEs) associated with direct targets of DAF-16/FOXO and DAF-16-associated elements (DAEs) associated with indirect targets may be differentially regulated by *daf-18(-)* and *daf-18(yh1)* in *daf-2(-)* mutants. Further research is required to test these possibilities for better understating of regulation of DAF-16/FOXO by DAF-

18/PTEN.”

- I recommend that the authors add a scheme at the end of the manuscript to make more clear their findings for potential readers which are not experts on lifespan regulation by DAF-16/SKN-1

David Vilchez

> We appreciate the reviewer’s comment. We added a schematic model as Fig. 6i.

Results, page 15, line 302: “Altogether, *daf-18(yh1)* appears to confer healthy longevity with minimal detrimental effects on the fitness of animals with reduced IIS by retaining DAF-16/FOXO activity while simultaneously dampening the harmful activation of SKN-1/NRF2 (Fig. 6i).”

Figure 6 legends, page 52, line 958: “(i) A schematic showing that *daf-18(yh1)* retains partial transcriptional activity of DAF-16/FOXO while suppressing the hyperactivation of PMK-1/p38 MAPK and SKN-1/NRF2 in *daf-2(-)* mutant animals, which contributes to healthy longevity. Thus, *daf-18(yh1)* increases healthy periods, while decreasing the time of frailty caused by longevity-promoting *daf-2(-)* mutations. A dotted, upward arrow indicates partial activity maintenance, and dotted, downward arrows indicate suppression of hyperactivation.”

Reviewer #2 (Remarks to the Author):

Loss of function mutations in the *C. elegans* insulin receptor (DAF-2) increases longevity and resistance to stressors such as oxidative stress and pathogens. However with this increased longevity and stress resistance, there are pleiotropic effects that cause a reduction in health span eg. low brood sizes, low feeding and motility. This manuscript aims to provide data to uncouple the association between longevity and reduced fitness seen in the reduction on of insulin signaling.

Here Park et al. report a novel hypomorphic *daf-18/pten* allele, *daf-18(yh1)*, that suppresses the healthspan defects in a hypomorphic *daf-2/insulin receptor* mutant *daf-2(e1370)*, but maintains its long lifespan and resistance to stressors such as oxidative stress and pathogenic bacteria (i.e. *Pseudomonas aeruginosa*).

The *daf-18(yh1)* allele was identified in an EMS mutagenesis screen for suppressors of the *daf-2(e1370)* temperature sensitive dauer constitutive phenotype. It is important to note that the *daf-2(e1370)* insulin receptor mutant is not a null allele and does retain some activity.

The *daf-18(yh1)* behaves differently from the null allele *daf-18(nr2037)* in that the *daf-18(-)* null in addition to suppressing the dauer constitutive phenotype, also suppresses the adult longevity, pathogen resistance.

Transcriptional analysis shows that the *daf-18(yh1)* behaves differently than the *daf-18* null in the *daf-2(-)* background, and in particular the *daf-18(yh1)* seems to have only a minor effect on the overactive DAF-16/FOXO which might explain why longevity is still maintained in the *daf-2(-); daf-18(yh1)* double. In this study, the author show that *daf-18(null)* increases SKN-1/NRF activity in *C. elegans* with reduced IIS (*daf-2(e1370)*), which contributes to shortened lifespan and health span, in contrast the *daf-2(-); daf-*

18(yh1) did not show this increased SKN-1/NRF activity and explains why longevity is retained and health span defects suppressed.

The daf-18(yh1) allele encodes a missense mutation in a conserved cysteine residue in the phosphatase domain (C150Y). The evidence provided suggests that daf-18(yh1) is a hypomorphic mutation, that is to say it retains some activity and not a null allele. Since DAF-18 like its human counterpart, PTEN, antagonizes IIS at the level of PI3K/AGE-1 then loss of daf-18 function leads to increased IIS. As I read the manuscript I could not help think that that the differences in phenotypes of daf-2(-); daf-18(yh1) versus daf-2(2-); daf-18(null) could simply be explained by the amount of IIS signaling in the daf-2(e1370) is just greater in the daf-18 null when compared to the daf-18(yh1)

To the authors credit, they do bring up the hypothesis that maybe discrete thresholds exist for the suppression of different IIS-regulated phenotypes by a weaker allele, daf-18(yh1), compared with a stronger allele, for example daf-18(-). daf-2(-) mutants may exhibit a lower threshold for the suppression of dauer formation than the enhanced pathogen resistance and longevity. To address this, they proposed that multiple hypomorphic mutations should also behave like daf-18(yh1) ie suppress dauer but not the PA14 resistance. They tested 6 alleles of daf-18 and all of them completely suppressed both dauer formation and PA14 resistance (i.e. behaved like the null). One exception was a missense mutant allele, daf-18(pe407) that suppressed the PA14 resistance but not the dauer constitutive daf-2(e1370) phenotype. It was not stated what the missense mutation is in the pe407 allele. This should be included.

> Following the reviewer's suggestion, we included more information regarding the nature of the *pe407* missense mutant allele in the Supplementary Figure 6 legends.

Supplementary Figure 6 legends, page 14, line 122: "*pe407* was isolated from the mutagenesis screen as a suppressor that restores the plasticity of salt chemotaxis in *casY-1(tm718)* mutant backgrounds¹⁰. *pe407* causes P140S change in the phosphatase domain of DAF-18/PTEN."

The authors conclude that because all 6 *daf-18* alleles, with the one exception, suppressed both phenotypes that it refutes the possibility that the threshold for the suppression of constitutive dauer phenotype in *daf-2(-)* mutants is lower than that of enhanced pathogen resistance and conclude the *yh1* allele appears to be a specific reduction-of-function *daf-18* allele for retaining immunity and longevity while suppressing dauer formation in *daf-2(-)* mutants.

I don't think they have ruled out the IIS threshold level hypothesis, and in fact the *daf-2(e1370)* allele itself shows temperature sensitivity for the dauer constitutive phenotype indicating that the IIS levels may be different at different temperatures supporting differences in threshold levels of IIS-regulated phenotypes. Since we do not know what is the functional consequences of the 6 *daf-18* alleles tested are , it is impossible to conclude that there are no IIS threshold levels for different IIS related processes. All 6 alleles tested could reduce DAF-18 function enough such that threshold levels of IIS are allowed to suppress the dauer, and PA resistant phenotypes.

> We agree with the reviewer's comment that we did not rule out the threshold

hypothesis, and therefore downplayed our statement in the manuscript by describing the limitation of our results as follows,

Results, page 11, line 219: “Despite the limitation due to the small number of tested *daf-18* mutant alleles, these data are not consistent with the possibility that the threshold for the suppression of constitutive dauer phenotype in *daf-2(-)* mutants is lower than that of enhanced pathogen resistance.”

How do these 6 *daf-18* alleles behave with respect to the *daf-2(e1370)*; DAF-16::GFP localization? Do they affect the nuclear localization of DAF-16::GFP in *daf-2(-)* mutants at an intermediate level between *daf-18(+)* and *daf-18(-)* (Fig. 4b,c). If the 6 alleles behave like *daf-18* null for DAF-16::GFP localization that could explain why PA14 resistance is suppressed with these *daf-18* alleles.

The DAF-16 and SKN-1 localization should be included for these alleles or some more phenotypic characterization order them in an allelic series.

> We agree with the reviewer’s comment that showing the effects of the six *daf-18* alleles on the subcellular localization of DAF-16 and SKN-1 will strengthen our points. Therefore, we prepared all the required strains, the six *daf-18* alleles that we tested with *daf-16::GFP* and *skn-1::GFP* transgenes in the *daf-2(e1370)* background by performing genetic crosses. We found that all the six tested *daf-18* alleles increased the nuclear localization of SKN-1::GFP in *daf-2(e1370)* animals to higher levels than *daf-18(yh1)* did (Supplementary Fig. 6e). These results suggest that *daf-18(yh1)* is a specific allele that did not substantially hyperactive SKN-1, unlike the other tested *daf-18* alleles. We also

found that three *daf-18* alleles, *e1375*, *mu398*, and *pe407*, retained the DAF-16::GFP nuclear localization with an extent similar to or higher than that of *yh1* in *daf-2* mutants (Supplementary Fig. 6d). Together these results suggest that *yh1* is a specific *daf-18* hypomorphic allele that confers beneficial physiological traits in *daf-2* mutants both by suppressing the hyperactivation of SKN-1 and by retaining partial activity of DAF-16 in *daf-2(-)* mutants. We described these results in the manuscript as follows,

Results, page 13, line 267: “We further examined the effects of the six additional *daf-18* alleles that we physiologically tested (Supplementary Fig. 6a-c) on the activities of DAF-16/FOXO and SKN-1/NRF2 by measuring their subcellular localization. We found that three *daf-18* alleles, *e1375*, *mu398*, and *pe407*, retained the DAF-16::GFP nuclear localization with an extent similar to or higher than that of *yh1* in *daf-2* mutants (Supplementary Fig. 6d). We then showed that all these six *daf-18* alleles increased the nuclear localization of SKN-1::GFP in *daf-2(e1370)* animals different from *daf-18(yh1)* (Supplementary Fig. 6e). These results indicate that *daf-18(yh1)* is a distinctive allele that did not hyperactivate SKN-1/NRF2, unlike all the other tested *daf-18* alleles. Together, these results suggest that *yh1* is a specific *daf-18* hypomorphic allele that confers beneficial physiological traits in *daf-2* mutants by limiting the hyper-activation of SKN-1/NRF2 and retaining partial activity of DAF-16/FOXO in *daf-2(-)* mutants.”

Supplementary Figure 6 legends, page 14, line 141: “(d,e) The effects of various *daf-18* mutant alleles on the subcellular localization of DAF-16::GFP and SKN-1::GFP in *daf-2(-)* mutants. (d) Quantification of the subcellular localization of DAF-

16::GFP in the intestines of indicated strains. Cytosolic: predominant cytosolic localization, intermediate: partial nuclear localization, nuclear: predominant nuclear localization (n ≥ 28 for each condition, from four to seven independent trials). (e) Quantification of the subcellular localization of SKN-1::GFP in the intestinal cells of indicated strains. Low: very dim GFP in the nuclei, medium: < 50% of the nuclei with SKN-1::GFP, high: > 50% of the nuclei with SKN-1::GFP (n ≥ 144 for each condition, from four to eight independent trials). The quantification data of DAF-16::GFP subcellular localization in WT, *daf-2(-)*, *daf-2(-); daf-18(yh1)*, and *daf-2(-); daf-18(-)* animals shown in panel d are the same experimental sets shown in Fig. 5g, and those in panel e are the same experimental sets shown in Fig. 5i. Error bars represent the standard error of mean (s.e.m., **p* < 0.05, ***p* < 0.01, ****p* < 0.001, n.s.: not significant, Chi-squared test).”

Genetics with *skn-1*gf alleles and *pmk-1* (upstream activator of SKN-1) suggested that hyperactivation of SKN-1/NRF contributes to the short lifespan of *daf-2(-); daf-18(-)* worms. In *daf-2(-); daf-18(yh1)* the hyperactivation of SKN-1 is not seen explaining why *daf-18(yh1)* (in addition to DAF-16 activation) does not suppress the longevity phenotype in the *daf-2(-)* mutants.

If this was the case why not test *skn-1* loss of function mutants or RNAi on *daf-2(-); daf-18(-)* animals to see if this increases the longevity?

> We thank the reviewer’s comment. We performed lifespan and health span (motility) assays with *skn-1* reduction-of-function mutants, *skn-1(zj15)* (Fig. 6c,g) [Please note that we used *skn-1(zj15)*, a point mutation that causes mis-splicing and reduces mRNA

levels of *skn-1* (Tang et al., 2015), because a strong *skn-1* mutant allele, *skn-1(mg570)*, causes sickness and short lifespan (Lehrbach and Ruvkun, 2016); indeed, we found that *mg570* substantially increased vulval rupture phenotypes in worms: 6% in wild-type and 22% in *daf-2(-); daf-18(nr2037)* backgrounds]. We found that *skn-1(zj15)* extended the short lifespan of *daf-2(e1370); daf-18(nr2037)* animals (Fig. 6c). In contrast, we found that *skn-1(zj15)* decreased the long lifespan of *daf-2(e1370); daf-18(yh1)* animals, indicating that retaining a proper SKN-1 activity in *daf-2(e1370); daf-18(yh1)* is required for longevity in these animals (Fig. 6c). These results suggest that long and healthy lifespan is sensitive to SKN-1 activity under reduced IIS conditions. We also measured motility (swimming) span of these worms but *skn-1(zj15)* did not affect either the motility of *daf-2(e1370); daf-18(nr2037)* or that of *daf-2(e1370); daf-18(yh1)* animals (Fig. 6g). The weak impact of *skn-1(zj15)* on the motility of worms may be due to a weak nature of the *zj15* allele, which may not affect the motility of worms in general. We included these results in the manuscript as follows,

Results, page 14, line 290: “Conversely, we found that a reduction-of-function allele *skn-1(zj15)* [*skn-1(-)*]⁴⁷ extended the short lifespan of *daf-2(-); daf-18(-)* animals, but not that of *daf-2(-); daf-18(yh1)* animals (Fig. 6c).”

Figure 6 legends, page 51, line 943: “(a-d) Effects of *daf-16(RNAi)* [*daf-16(-)*] (a), *skn-1(lax188)* [*skn-1(gf)*] (b), *skn-1(zj15)* [*skn-1(-)*] (c), and *pmk-1(RNAi)* [*pmk-1(-)*] (d) on the lifespan of *daf-2(e1370); daf-18(yh1)* [*daf-2(-); daf-18(yh1)*] and *daf-2(-); daf-18(nr2037)* [*daf-18(-)*] animals (n ≥ 240 for each condition). All the lifespan assays were performed at least twice. (e-h) Effects of *daf-16(-)* (e), *skn-1(gf)* (f),

skn-1(-) (**g**), and *pmk-1(-)* (**h**) on the swimming rate (motility) of *daf-2(-); daf-18(yh1)* and *daf-2(-); daf-18(-)* animals at day 0 and day 7 adulthoods (n = 30 for each condition, from three independent trials). See Supplementary Figure 10c for the effects of *skn-1(-)* on the lifespan of *daf-2(-)* and wild-type (WT) animals. See Supplementary Figure 10e,f for the requirement of PMK-1 for the decreased PA14 susceptibility of *daf-2(-)* and *daf-2(-); daf-18(yh1)* worms and for the normal survival of WT on PA14. We found that *skn-1(-)* did not affect either the motility of *daf-2(-); daf-18(-)* or that of *daf-2(-); daf-18(yh1)* worms in **g**, and this may be due to the weak nature of the *zj15* allele, which needs to be tested in various other genetic backgrounds in future research.”

Supplementary Figure 10 legends, page 22, line 199: “**(a-c)** Effects of *daf-16* RNAi [*daf-16(-)*] (n = 120 for each condition) (**a**), *skn-1(lax188)* [*skn-1(gf)*] (n ≥ 295 for each condition) (**b**), *skn-1(zj15)* [*skn-1(-)*] (n ≥ 270 for each condition) (**c**), and *pmk-1* RNAi [*pmk-1(-)*] (n = 120 for each condition) (**d**) on the lifespan of wild-type (WT) and *daf-2(e1370)* [*daf-2(-)*] worms. Please note that we used *skn-1(zj15)*, a point mutation that causes mis-splicing and reduces the mRNA levels of *skn-1*, because a strong *skn-1* mutant allele, *skn-1(mg570)*, causes sickness and short lifespan²⁸; indeed, we found that *mg570* substantially increased vulval rupture phenotypes in worms: 6% in WT and 22% in *daf-2(-); daf-18(nr2037)* [*daf-18(-)*] backgrounds.”

What are the phenotypes of the *daf-18(yh1)* alone? It is mentioned that alone they have reduced healthspan, but the authors should show the data on the *daf-18(yh1)* single

mutant compared to the *daf-18*(null) on longevity and, dauer formation. How does *daf-18*(*yh1*) alone compare to a *daf-18* null for these classic IIS phenotypes is important information and will support the hypomorphic nature of the *yh1* allele.

> We appreciate the reviewer for raising this issue. We described the results regarding survival and health span assays using *daf-18*(*yh1*) single mutants in the manuscript (Supplementary Fig. 2). We found that *daf-18*(*yh1*) decreased lifespan, swimming span, and feeding span in the wild-type background with extents similar to that caused by *daf-18*(*nr2037*) at 20°C (Supplementary Fig. 2c,f,g). We also determined the effects of *daf-18*(*yh1*) on pathogen resistance (Supplementary Fig. 2d) and dauer formation at 27°C in the wild-type background (Supplementary Fig. 2e). We found that *daf-18*(*yh1*) mutation did not affect the survival of wild-type animals upon PA14 infection and substantially suppressed dauer formation at 27°C. We described these results as follows,

Supplementary Information, page 4, line 36: “*daf-18*(*yh1*) substantially decreased the lifespan of WT animals with an extent similar to that caused by *daf-18*(-) at 20°C (four out of six trials). The lifespan curves of WT, *daf-2*(-), *daf-2*(-); *daf-18*(*yh1*), and *daf-2*(-); *daf-18*(-) shown in panel **a** are the same experimental sets shown in Fig. 1i, and those in **c** are the same experimental sets shown in Supplementary Fig. 10b and Fig. 6b. **(d-g)** The effects of *daf-18*(*yh1*) and *daf-18*(-) on pathogen resistance (n = 180 for each condition) **(d)**, dauer formation at 27°C (n ≥ 790 for each condition, from at least eight trials) **(e)**, swimming span (n = 10 for each condition, from one trial) **(f)**, and feeding span (n = 10 for each condition, from one trial) **(g)**. The survival of *daf-18*(*yh1*) animals under PA14 infection was similar to that of WT worms, whereas *daf-18*(-) worms displayed reduced survival **(d)**. Both *daf-18*(*yh1*) (0.3%)

and *daf-18(-)* (0%) mutations substantially suppressed dauer phenotypes at 27°C; WT animals displayed 7.1% of dauers at 27°C, and *tax-4(p678)* [*tax-4(-)*] animals, which display a constitutive dauer phenotype at 27°C², was used as a positive control (e). In addition, *daf-18(yh1)* and *daf-18(-)* mutations similarly reduced the swimming (motility) and feeding spans (f,g). Error bars indicate the standard error of mean (s.e.m., ****p* < 0.001, two-tailed Student's *t*-test relative to WT)."

Methods, page 27, line 580: "Dauer assays were performed as previously described²⁰, with minor modifications. Gravid adult worms were allowed to lay eggs on NGM plates at 25°C or 27°C depending on assay conditions and removed after three to six hrs for synchronization of eggs. The F₁ progeny were examined for dauer formation after three or four days at 25°C or 27°C. Dauer formation was visually determined¹⁴ under a dissecting stereomicroscope (SMZ645, Nikon, Tokyo, Japan)."

Only one *daf-2* allele was used in this study (e1370). Does *daf-18(yh1)* suppress the *daf-2* null (ie *daf-2(e979)*) for longevity? What about other *daf-2* alleles eg. The alleles that affect the receptor binding domain?

> We thank the reviewer's comment. We performed lifespan and PA14 resistance assays using ligand-binding domain-defective *daf-2(e979)* mutants as the reviewer suggested (Supplementary Fig. 3a-f) (Gems et al., 1998; Patel et al., 2008). Consistent with survival assay results using *daf-2(e1370)* mutants, we found that *daf-18(yh1)* partially retained longevity and enhanced pathogen resistance in *daf-2(e979)* animals

(Supplementary Fig. 3e,f). We also obtained similar results using *daf-2* RNAi (Supplementary Fig. 3a,b). However, we found that *daf-18(yh1)* did not retain the longevity and enhanced pathogen resistance in *daf-2(e1368)* mutants (Supplementary Fig. 3c,d), the class 1 *daf-2* mutants with weaker longevity and stress resistance phenotypes than *daf-2(e1370)* and *daf-2(RNAi)* (Gems et al., 1998; Patel et al., 2008; Ewald et al., 2015). Together, these results suggest that *daf-18(yh1)* can maintain longevity and pathogen resistance in animals with reduced IIS in multiple situations and that the effect of *daf-18(yh1)* on these phenotypes is not specific for *daf-2(e1370)*. In addition, we propose that a certain level of IIS reduction is required for *daf-18(yh1)* to elicit beneficial effects on healthy longevity. However, one limitation of our speculation is that testing four alleles of *daf-2* is not sufficient to conclude anything very strong. Therefore, we briefly mentioned these new results in the manuscript as follows,

Results, page 7, line 130: “We then tested the effects of *daf-18(yh1)* on lifespan and PA14 resistance conferred by other mutations in *daf-2*, a weak *daf-2(e1368)* allele and a strong ligand-binding domain-defective *daf-2(e979)* allele, by RNAi knockdown of *daf-2*^{21,22}, and by a hypomorphic *hx546* allele of *age-1*, which encodes phosphoinositide 3-kinase that counteracts DAF-18/PTEN phosphatase^{5,6}. We found that *daf-18(yh1)* partially maintained longevity and enhanced immunity in *daf-2(RNAi)* and *daf-2(e979)* worms (Supplementary Fig. 3a,b,e,f), but did not in *daf-2(e1368)* or *age-1(hx546)* mutants (Supplementary Fig. 3c,d,g,h). These results suggest that *daf-18(yh1)* can maintain lifespan and pathogen resistance in animals with reduced IIS caused by genetically inhibited *daf-2* with multiple intervention modes, not specific to *daf-2(e1370)*. In addition, because *daf-2(e1368)* and *age-*

1(hx546) mutations cause weak longevity and stress resistance phenotypes (Supplementary Fig. 3c,d,g-j)²¹⁻²⁵, we propose that a certain level of IIS reduction is required for *daf-18(yh1)* to maintain longevity and immunity.”

Supplementary Figure 3 legends, page 6, line 55: **“Supplementary Figure 3: *daf-18(yh1)* retains lifespan and pathogen resistance in worms with reduced IIS in multiple conditions. (a-h)** The effects of *daf-18(yh1)* and *daf-18(nr2037)* [*daf-18(-)*] on lifespan at 20°C (n ≥ 270 for each condition; n = 156 for *daf-2(e979)* animals) and pathogen resistance (n = 180 for each condition; n = 35 for *daf-2(e979)* animals) of *daf-2(RNAi)* (a,b), *daf-2(e1368)* (c,d), *daf-2(e979)* (e,f), and *age-1(hx546)* [*age-1(-)*] (g,h) animals. (i,j) Swimming rate (n = 30 for each condition, body bends per minute in liquid measured from three independent trials) of day 0 (i) and day 7 adult (j) wild-type (WT), *age-1(-); daf-18(+)*, *age-1(-); daf-18(yh1)*, and *age-1(-); daf-18(-)* animals. Error bars indicate the standard error of mean (s.e.m., **p* < 0.05, ***p* < 0.01, n.s.: not significant, two-tailed Student’s *t*-test relative to WT unless otherwise noted). See Supplementary Tables 1 and 4 for additional repeats and statistical analysis for the survival assay and health span assay data shown in this figure.”

DAF-18/PTEN antagonizes the the PI3k/AGE-1 PI kinase does *daf-18(yh1)* behave differently than *daf-18(null)* with respect to suppression of *age-1* loss of function phenotypes or is the the suppressors phenotypes unique to *daf-2*?

> We thank the reviewer’s comment. To address whether *daf-18(yh1)* behaves differently from *daf-18(nr2037)* in *age-1* loss-of-function mutants, we performed lifespan,

PA14 resistance, and health span (motility span) assays using long-lived, standard hypomorphic *age-1(hx546)* mutants. We found that *daf-18(yh1)* suppressed longevity and enhanced immunity in *age-1(hx546)* animals (Supplementary Fig. 3g,h). AGE-1/PI3K is a lipid kinase and counteracts DAF-18/PTEN phosphatase. Therefore, both *daf-18(yh1)* and *daf-18(nr2037)* that substantially reduced the lipid phosphatase activity of DAF-18/PTEN may have canceled out the beneficial effects of *age-1* mutation on lifespan and immunity. We also found that *age-1(hx546)* mutation had no significant effect on the motility of worms (Supplementary Fig. 3i,j). We therefore speculate that *hx546* had small or no effects on the motility of worms because of a weak nature of *hx546* allele (Friedman and Johnson, 1988; Ayyadevara et al., 2009). One caveat for this speculation is that we only used one *age-1* mutant allele for the revision, and therefore we cannot conclude anything strong. Therefore, we included these results briefly in the manuscript as follows,

Results, page 7, line 130: “We then tested the effects of *daf-18(yh1)* on lifespan and PA14 resistance conferred by other mutations in *daf-2*, a weak *daf-2(e1368)* allele and a strong ligand-binding domain-defective *daf-2(e979)* allele, by RNAi knockdown of *daf-2*^{21,22}, and by a hypomorphic *hx546* allele of *age-1*, which encodes phosphoinositide 3-kinase that counteracts DAF-18/PTEN phosphatase^{5,6}. We found that *daf-18(yh1)* partially maintained longevity and enhanced immunity in *daf-2(RNAi)* and *daf-2(e979)* worms (Supplementary Fig. 3a,b,e,f), but did not in *daf-2(e1368)* or *age-1(hx546)* mutants (Supplementary Fig. 3c,d,g,h). These results suggest that *daf-18(yh1)* can maintain lifespan and pathogen resistance in animals with reduced IIS caused by genetically inhibited *daf-2* with multiple intervention

modes, not specific to *daf-2(e1370)*. In addition, because *daf-2(e1368)* and *age-1(hx546)* mutations cause weak longevity and stress resistance phenotypes (Supplementary Fig. 3c,d,g-j)²¹⁻²⁵, we propose that a certain level of IIS reduction is required for *daf-18(yh1)* to maintain longevity and immunity.”

Supplementary Figure 3 legends, page 6, line 55: “**Supplementary Figure 3: *daf-18(yh1)* retains lifespan and pathogen resistance in worms with reduced IIS in multiple conditions.** (a-h) The effects of *daf-18(yh1)* and *daf-18(nr2037)* [*daf-18(-)*] on lifespan at 20°C (n ≥ 270 for each condition; n = 156 for *daf-2(e979)* animals) and pathogen resistance (n = 180 for each condition; n = 35 for *daf-2(e979)* animals) of *daf-2(RNAi)* (a,b), *daf-2(e1368)* (c,d), *daf-2(e979)* (e,f), and *age-1(hx546)* [*age-1(-)*] (g,h) animals. (i,j) Swimming rate (n = 30 for each condition, body bends per minute in liquid measured from three independent trials) of day 0 (i) and day 7 adult (j) wild-type (WT), *age-1(-); daf-18(+)*, *age-1(-); daf-18(yh1)*, and *age-1(-); daf-18(-)* animals. Error bars indicate the standard error of mean (s.e.m., **p* < 0.05, ***p* < 0.01, n.s.: not significant, two-tailed Student’s *t*-test relative to WT unless otherwise noted). See Supplementary Tables 1 and 4 for additional repeats and statistical analysis for the survival assay and health span assay data shown in this figure.”

The novel finding in this manuscript is that the authors have identified a point mutation in DAF-18/PTEN that uncouples longevity from impaired fitness in *C. elegans* with reduced IIS.

The *daf-18(yh1)* changes a conserved cysteine to a tyrosine in the phosphatase domain

(C150Y). The authors propose that the *yh1* is a specific reduction of function *daf-18* allele that allows the DAF-18 protein to retain pathogen immunity and longevity, while suppressing dauer formation in *daf-2(-)* mutants.

However this could still be argued that the *yh1* is a hypomorph and that it retains some function and thus still has the ability to reduce the IIS that is active in the *daf-2(e1370)*, thereby changing the IIS levels in the *daf-2 (e1370)* mutant. This is consistent with the threshold level of IIS regulating different processes.

Where this manuscript falls short is that if the C150Y is really a special mutation, then how does this mutation give DAF-18/PTEN its special properties? Figure 3a-c show that the C150Y does indeed affect DAF-18/PTEN lipid phosphatase activity. But when you compare the *yh1* to the *daf-18* null they both reduce the lipid phosphatase activity to similar levels and not significantly different. The analogous human PTEN C105Y version was created but it also shows reduction in lipid phosphatase activity to the well-studied PTEN C124S phosphatase-dead variant. So the C150Y behaves similar to a null for lipid phosphatase activity. This questions whether DAF-18 C150Y has lipid phosphatase independent activity. Note that feeding and brood size suppression is not significantly different than the *daf-18* null and suggests that *yh1* reduces DAF-18 function enough to allow *daf-2(e1370)* IIS to suppress these phenotypes.

Since PTEN has also been shown to have protein phosphatase and important question to address is whether the DAF-18 C150Y or PTEN C105Y retains this protein phosphatase activity? This would make the case that *yh1* is a special loss of function.

> We appreciate the reviewer for raising this important issue. To test whether PTEN^{C105Y} retained protein phosphatase activity, we performed protein phosphatase assays using

a synthesized generic peptide harboring phospho-tyrosine, because PTEN has been shown to act as a protein tyrosine phosphatase (Myers et al 1997; Zhang et al 2012). We found that PTEN^{C105Y} retained the protein phosphatase activity partially but substantially (57.3%) compared with wild-type DAF-18/PTEN, whereas a negative control, PTEN^{C124S}, dramatically decreased that (Fig. 3g). We further measured dose-dependent activity changes of the PTEN variants to assess the kinetics of protein phosphatase activity (Fig. 3h). For this assay, we included two controls, the phosphatase-dead variant PTEN^{C124S} as a negative control, and protein tyrosine phosphatase 1B (PTP1B) as a positive control (Fig. 3h and Supplementary Fig. 4f). We showed that the PTEN^{C105Y} partially retained the protein phosphatase activity at various concentrations, and the activity of PTEN^{C105Y} was saturated with 37% compared with the total activity of wild-type DAF-18/PTEN, whereas PTEN^{C124S} barely exhibited an enzyme activity even at the highest concentration. Thus, our data indicate that *daf-18(yh1)* retains the partial protein phosphatase activity of DAF-18/PTEN. These results raise the possibility that the remaining protein phosphatase activity of DAF-18^{C150Y} contributes to longevity and enhanced immunity in *daf-2(-)* animals. We added the results to the revised manuscript as follows,

Abstract, page 2, line 27: “We showed that the *daf-18(yh1)* mutation decreased the lipid phosphatase activity of DAF-18/PTEN, while retaining a partial protein tyrosine phosphatase activity.”

Introduction, page 5, line 79: “We found that *daf-18(yh1)* substantially decreased the lipid phosphatase activity of DAF-18/PTEN, while partly maintaining its protein

phosphatase activity.”

Results, Page 9, line 171: “Because PTEN also acts as a protein tyrosine phosphatase^{27,28}, we performed protein phosphatase assays using a synthesized generic peptide harboring phospho-tyrosine. We found that PTEN^{C105Y} retained a substantial protein phosphatase activity (57.3%) compared with wild-type DAF-18/PTEN, whereas the phosphatase-dead PTEN^{C124S} (negative control) dramatically decreased that (Fig. 3g). We confirmed the results by measuring dose-dependent changes of the tyrosine phosphatase activity (Fig. 3h and Supplementary Fig. 4f). Thus, *daf-18(yh1)* appears to retain the partial tyrosine phosphatase activity of DAF-18/PTEN, raising the possibility that the protein phosphatase activity of DAF-18^{C150Y} contributes to longevity and enhanced immunity in *daf-2(-)* animals.”

Figure 3 legends, page 44, line 859: “**Figure 3: *daf-18(yh1)* substantially decreases lipid phosphatase activity, but partially maintains protein phosphatase activity of DAF-18/PTEN.** (a) Representative images of *rpl-28p::CFP::PH_{AKT}*-expressing worms with *daf-2(e1370)* [*daf-2(-)*], *daf-2(-); daf-18(yh1)*, or *daf-2(-); daf-18(nr2037)* [*daf-18(-)*] mutations (*rpl-28p*: a promoter of a ubiquitous *rpl-28*, ribosomal protein large subunit 28). Scale bar: 50 μ m. Arrowhead: membrane CFP::PH_{AKT}. (b) Quantification of membrane-localized PH_{AKT} in worms in panel a (n \geq 23 for each condition, from four independent trials). (c,d) His-tagged human recombinant PTEN proteins used for *in vitro* phosphatase assay. WT: wild-type PTEN; C105Y: C105Y mutant PTEN; C124S: C124S mutant PTEN (phosphatase-dead variant). The recombinant proteins were separated by using

SDS-PAGE and stained with Coomassie blue (c), and detected by using western blotting with anti-His antibody (d). (e,f) *In vitro* PTEN lipid phosphatase assay. Purified recombinant WT, C105Y, and C124S PTEN proteins [127 nM (e) or 42 nM (f)] were incubated with PIP₃ substrates (N = 4). (g) *In vitro* PTEN protein tyrosine phosphatase assay. Purified recombinant WT, C105Y, and C124S PTEN proteins were incubated with phospho-tyrosine-containing peptides, and the protein phosphatase activities were calculated by detecting free phosphates (N = 9). (h) Dose-dependent changes in protein phosphatase activities of the PTEN variants (N = 3). See Supplementary Fig. 4f for the comparison of protein phosphatase activities between WT PTEN and protein tyrosine phosphatase 1B (PTP1B), a positive control. N.D: not detected. Error bars represent the standard error of mean (s.e.m., *** $p < 0.001$, n.s.: not significant, two-tailed Student's *t*-test relative to WT unless otherwise noted).”

Supplementary Figure 4 legends, page 9, line 78: “(f) Dose-dependent changes in the protein phosphatase activity of human recombinant PTEN (WT) and protein tyrosine phosphatase 1B (PTP1B, positive control). The indicated amounts of recombinant proteins were tested for the protein phosphatase assay (N = 3). Note that the same dose-dependent protein phosphatase activity curve of WT is shown in Fig. 3h as well.”

Discussion, page 15, line 315: “Notably, our data revealed that a specific mutation in *daf-18/PTEN* preserved the partial protein phosphatase activity of DAF-18/PTEN and transcriptional activity of DAF-16/FOXO while preventing the harmful activation

of transcription factor SKN-1/NRF2, leading to the differential physiological effects.”

Methods, page 34, line 729: “*In vitro* lipid and protein phosphatase assays were performed with a malachite green phosphatase assay kit (K-1500, Echelon Biosciences, Salt Lake City, UT, USA) following the manufacturer’s instruction. For lipid phosphatase assays, soluble phosphatidylinositol 3,4,5-trisphosphate diC8 (PIP₃ diC8, Echelon Biosciences, Salt Lake City, UT, USA) was diluted to 1 mM in distilled water. Indicated amounts of purified PTEN recombinant proteins were incubated with 3 µl of the 1 mM PIP₃ solution in 25 µl Tris-buffer [25 mM Tris-HCl (pH 7.4), 140 mM NaCl, 2.7 mM KCl and 10 mM DTT] for 40 min at 37°C. For protein tyrosine phosphatase assays, a synthetic phospho-tyrosine peptide (YEEEEpYEEEE) was used as a substrate⁷⁵. Indicated amounts of recombinant His-PTEN proteins (WT, C105Y and C124S) and the protein tyrosine phosphatase 1B (PTP1B: positive control, R&D system, Minneapolis, MN, USA) were used as enzymes. The peptide substrates (100 µM) were incubated with each enzyme in 25 µl reaction buffer [50 mM Tris-HCl (pH 7.4), 10 mM DTT] for 60 min at 37°C. One hundred µl of malachite green solution was then added to terminate the enzyme reaction and incubated for 20 min at room temperature. The released phosphate was quantified by measuring absorption spectrum at 620 nm using a microplate reader. A standard curve derived from 0.1 mM phosphate provided by the assay kit was used to convert the absorbance value at 620 nm to amount of free phosphate. To obtain dose-response curves of PTEN enzymes, the protein concentration was increased stepwise by 2-fold to reach a final 10 nM concentration for the protein phosphatase assay. The dose-dependent activity was not observed for the

phosphatase activity-dead variant, PTEN^{C124S}.”

The last paragraph has some bold statements such as suggesting “the introduction of a specific yh1-like mutation in PTEN to mammalian mutants defective in IIS may boost fitness.” This section should be toned down, as the analogous PTEN Y105C on its own in humans has been associated with sporadic Bannayan-Riley-Ruvalcaba Syndrome.

> We appreciate the reviewer’s comment. We agree with the reviewer and removed the statement from the text (Discussion, page 18, line 381).

Reviewer #3 (Remarks to the Author):

In this article, Park et al., describe an allelic variant of the *C. elegans* gene *daf-18* that uncouples improved longevity and immunity from developmental defects observed in mutants of the insulin/IGF1 receptor, DAF-2. *daf-18* encodes the worm homolog of the tumor suppressor PTEN phosphatase. Unlike strong *daf-18* loss of function/null mutants, that abolishes *daf-2* longevity and immunity as well as developmental defects, the authors identified a new *daf-18* allele, *yh1*, that minimally effects longevity and immunity but protects the mutants from many of the other defects in developmental fitness measures. The allele is mapped to the phosphatase domain of the protein and the authors’ data suggest that it reduces phosphatase activity. Based on gene expression profiling and molecular genetic studies, the authors suggest that *yh1* produces the benefits it does by impacting the activities of two of the transcription factors employed in *daf-2* mutants longevity- by ‘maintaining’ the activity of the FOXO protein DAF-16 and

'preventing harmful activation' of the NRF2 homolog, SKN-1. The study focuses on a highly topical and important area, the molecular distinctions between lifespan and individual aspects of health and fitness, that is likely to be of great interest to readers in the aging field.

The work is based on highly interesting observations, the experiments are well designed, logical and conducted with rigor. One concern is that while the data is compelling in documenting the selective impact of the *daf-18(yh1)* mutation on longevity/immunity vs. developmental phenotypes, there is limited mechanistic insight into how and why this occurs (especially #1, 2 below) and the evidence supporting some key conclusions is rather weak.

1. How the *yh1* mutation different/impacting DAF-18 function as compared to the null/other alleles? The data with the PH-domain reporter and human PTEN is very encouraging in implicating altered phosphatase activity. But, both the null and *yh1* show similar effects in this assay. Other mutations that completely suppress dauer formation and immunity also map to the phosphatase domain (*ok480*) as does an allele that selectively/largely suppresses immunity alone (*pe407*). Since the *yh1* and CRISPR alleles do show modest reduction in immunity, and in the absence of data comparing/characterizing the impacts of the different alleles on phosphatase activity, the possibility that *yh1* is simply a weaker allele that crosses the threshold for dauer formation but not immunity cannot be overruled.

> We thank the reviewer's comment. To test whether *daf-18(yh1)* is a special *daf-18*

mutant allele, we performed protein phosphatase assays using a synthesized generic peptide harboring phospho-tyrosine, because PTEN has been shown to act as a protein tyrosine phosphatase (Myers et al 1997; Zhang et al 2012). We found that PTEN^{C105Y} retained the protein phosphatase activity partially but substantially (57.3%) compared with wild-type DAF-18/PTEN, whereas a negative control, PTEN^{C124S}, dramatically decreased that (Fig. 3g). We further measured dose-dependent activity changes of the PTEN variants to assess the kinetics of protein phosphatase activity (Fig. 3h). For this assay, we included two controls, the phosphatase-dead variant PTEN^{C124S} as a negative control, and protein tyrosine phosphatase 1B (PTP1B) as a positive control (Fig. 3h and Supplementary Fig. 4f). We showed that the PTEN^{C105Y} partially retained the protein phosphatase activity at various concentrations, and the activity of PTEN^{C105Y} was saturated with 37% compared with the total activity of wild-type DAF-18/PTEN, whereas PTEN^{C124S} barely exhibited an enzyme activity even at the highest concentration. Our data indicate that *daf-18(yh1)* retains the partial protein phosphatase activity of DAF-18/PTEN. These results raise the possibility that the remaining protein phosphatase activity of DAF-18/PTEN in the animals with *daf-18(yh1)* appears to allow *daf-2* mutants to retain longevity and health.

We also measured the subcellular localization of DAF-16::GFP and SKN-1::GFP in *daf-2(e1370)* mutants in combination with six additional *daf-18* mutant alleles that we functionally tested (Supplementary Fig. 6d,e). We found that all the six tested *daf-18* alleles increased the nuclear localization of SKN-1::GFP in *daf-2(e1370)* animals different from *daf-18(yh1)*, suggesting harmful effects of SKN-1 hyperactivation on immunity in these animals (Supplementary Fig. 6e). These results suggest that *daf-*

18(yh1) is a specific allele that did not substantially hyperactivate SKN-1, unlike the other tested *daf-18* alleles. We also found that three *daf-18* alleles, *e1375*, *mu398*, and *pe407*, retained the DAF-16::GFP nuclear localization with an extent similar to or higher than that of *yh1* in *daf-2* mutants (Supplementary Fig. 6d). Together these results suggest that *yh1* is a specific *daf-18* hypomorphic allele that confers beneficial physiological traits in *daf-2* mutants both by suppressing the hyper-activation of SKN-1 and by retaining partial activity of DAF-16 in *daf-2(-)* mutants. We described these results in the manuscript as follows,

Abstract, page 2, line 27: “We showed that the *daf-18(yh1)* mutation decreased the lipid phosphatase activity of DAF-18/PTEN, while retaining a partial protein tyrosine phosphatase activity.”

Introduction, page 5, line 79: “We found that *daf-18(yh1)* substantially decreased the lipid phosphatase activity of DAF-18/PTEN, while partly maintaining its protein phosphatase activity.”

Results, Page 9, line 171: “Because PTEN also acts as a protein tyrosine phosphatase^{27,28}, we performed protein phosphatase assays using a synthesized generic peptide harboring phospho-tyrosine. We found that PTEN^{C105Y} retained a substantial protein phosphatase activity (57.3%) compared with wild-type DAF-18/PTEN, whereas the phosphatase-dead PTEN^{C124S} (negative control) dramatically decreased that (Fig. 3g). We confirmed the results by measuring dose-dependent changes of the tyrosine phosphatase activity (Fig. 3h and Supplementary Fig. 4f). Thus, *daf-18(yh1)* appears to retain the partial tyrosine phosphatase activity of DAF-

18/PTEN, raising the possibility that the protein phosphatase activity of DAF-18^{C150Y} contributes to longevity and enhanced immunity in *daf-2(-)* animals.”

Figure 3 legends, page 44, line 859: “**Figure. 3: *daf-18(yh1)* substantially decreases lipid phosphatase activity, but partially maintains protein**

phosphatase activity of DAF-18/PTEN. (a) Representative images of *rpl-*

28p::CFP::PH_{AKT}-expressing worms with *daf-2(e1370)* [*daf-2(-)*], *daf-2(-)*; *daf-*

18(yh1), or *daf-2(-)*; *daf-18(nr2037)* [*daf-18(-)*] mutations (*rpl-28p*: a promoter of a

ubiquitous *rpl-28*, ribosomal protein large subunit 28). Scale bar: 50 μ m. Arrowhead:

membrane CFP::PH_{AKT}. (b) Quantification of membrane-localized PH_{AKT} in worms in

panel a (n \geq 23 for each condition, from four independent trials). (c,d) His-tagged

human recombinant PTEN proteins used for *in vitro* phosphatase assay. WT: wild-

type PTEN; C105Y: C105Y mutant PTEN; C124S: C124S mutant PTEN

(phosphatase-dead variant). The recombinant proteins were separated by using

SDS-PAGE and stained with Coomassie blue (c), and detected by using western

blotting with anti-His antibody (d). (e,f) *In vitro* PTEN lipid phosphatase assay.

Purified recombinant WT, C105Y, and C124S PTEN proteins [127 nM (e) or 42 nM

(f)] were incubated with PIP₃ substrates (N = 4). (g) *In vitro* PTEN protein tyrosine

phosphatase assay. Purified recombinant WT, C105Y, and C124S PTEN proteins

were incubated with phospho-tyrosine-containing peptides, and the protein

phosphatase activities were calculated by detecting free phosphates (N = 9). (h)

Dose-dependent changes in protein phosphatase activities of the PTEN variants (N

= 3). See Supplementary Fig. 4f for the comparison of protein phosphatase activities

between WT PTEN and protein tyrosine phosphatase 1B (PTP1B), a positive control.

N.D: not detected. Error bars represent the standard error of mean (s.e.m., *** $p < 0.001$, n.s.: not significant, two-tailed Student's t -test relative to WT unless otherwise noted).”

Supplementary Figure 4 legends, page 9, line 78: “(f) Dose-dependent changes in the protein phosphatase activity of human recombinant PTEN (WT) and protein tyrosine phosphatase 1B (PTP1B, positive control). The indicated amounts of recombinant proteins were tested for the protein phosphatase assay (N = 3). Note that the same dose-dependent protein phosphatase activity curve of WT is shown in Fig. 3h as well.”

Discussion, page 15, line 315: “Notably, our data revealed that a specific mutation in *daf-18/PTEN* preserved the partial protein phosphatase activity of DAF-18/PTEN and transcriptional activity of DAF-16/FOXO while preventing the harmful activation of transcription factor SKN-1/NRF2, leading to the differential physiological effects.”

Methods, page 34, line 729: “*In vitro* lipid and protein phosphatase assays were performed with a malachite green phosphatase assay kit (K-1500, Echelon Biosciences, Salt Lake City, UT, USA) following the manufacturer’s instruction. For lipid phosphatase assays, soluble phosphatidylinositol 3,4,5-trisphosphate diC8 (PIP₃ diC8, Echelon Biosciences, Salt Lake City, UT, USA) was diluted to 1 mM in distilled water. Indicated amounts of purified PTEN recombinant proteins were incubated with 3 μ l of the 1 mM PIP₃ solution in 25 μ l Tris-buffer [25 mM Tris-HCl (pH 7.4), 140 mM NaCl, 2.7 mM KCl and 10 mM DTT] for 40 min at 37°C. For protein tyrosine phosphatase assays, a synthetic phospho-tyrosine peptide

(YEEEEpYEEEE) was used as a substrate⁷⁵. Indicated amounts of recombinant His-PTEN proteins (WT, C105Y and C124S) and the protein tyrosine phosphatase 1B (PTP1B: positive control, R&D system, Minneapolis, MN, USA) were used as enzymes. The peptide substrates (100 μ M) were incubated with each enzyme in 25 μ l reaction buffer [50 mM Tris-HCl (pH 7.4), 10 mM DTT] for 60 min at 37°C. One hundred μ l of malachite green solution was then added to terminate the enzyme reaction and incubated for 20 min at room temperature. The released phosphate was quantified by measuring absorption spectrum at 620 nm using a microplate reader. A standard curve derived from 0.1 mM phosphate provided by the assay kit was used to convert the absorbance value at 620 nm to amount of free phosphate. To obtain dose-response curves of PTEN enzymes, the protein concentration was increased stepwise by 2-fold to reach a final 10 nM concentration for the protein phosphatase assay. The dose-dependent activity was not observed for the phosphatase activity-dead variant, PTEN^{C124S}.”

Results, page 13, line 267: “We further examined the effects of the six additional *daf-18* alleles that we physiologically tested (Supplementary Fig. 6a-c) on the activities of DAF-16/FOXO and SKN-1/NRF2 by measuring their subcellular localization. We found that three *daf-18* alleles, *e1375*, *mu398*, and *pe407*, retained the DAF-16::GFP nuclear localization with an extent similar to or higher than that of *yh1* in *daf-2* mutants (Supplementary Fig. 6d). We then showed that all these six *daf-18* alleles increased the nuclear localization of SKN-1::GFP in *daf-2(e1370)* animals different from *daf-18(yh1)* (Supplementary Fig. 6e). These results indicate that *daf-18(yh1)* is a distinctive allele that did not hyperactive SKN-1/NRF2, unlike

all the other tested *daf-18* alleles. Together, these results suggest that *yh1* is a specific *daf-18* hypomorphic allele that confers beneficial physiological traits in *daf-2* mutants by limiting the hyper-activation of SKN-1/NRF2 and retaining partial activity of DAF-16/FOXO in *daf-2(-)* mutants.”

Supplementary Figure 6 legends, page 14, line 141: “(d,e) The effects of various *daf-18* mutant alleles on the subcellular localization of DAF-16::GFP and SKN-1::GFP in *daf-2(-)* mutants. (d) Quantification of the subcellular localization of DAF-16::GFP in the intestines of indicated strains. Cytosolic: predominant cytosolic localization, intermediate: partial nuclear localization, nuclear: predominant nuclear localization (n ≥ 28 for each condition, from four to seven independent trials). (e) Quantification of the subcellular localization of SKN-1::GFP in the intestinal cells of indicated strains. Low: very dim GFP in the nuclei, medium: < 50% of the nuclei with SKN-1::GFP, high: > 50% of the nuclei with SKN-1::GFP (n ≥ 144 for each condition, from four to eight independent trials). The quantification data of DAF-16::GFP subcellular localization in WT, *daf-2(-)*, *daf-2(-); daf-18(yh1)*, and *daf-2(-); daf-18(-)* animals shown in panel d are the same experimental sets shown in Fig. 5g, and those in panel e are the same experimental sets shown in Fig. 5i. Error bars represent the standard error of mean (s.e.m., **p* < 0.05, ***p* < 0.01, ****p* < 0.001, n.s.: not significant, Chi-squared test).”

2. The evidence supporting the authors’ conclusion that *yh1* ‘retains DAF-16 activity but restricts detrimental SKN-1 upregulation’ is weak in premise and experimental strength

and described in a very confusing manner.

The comparisons of DAF-16-, SKN-1- mediated transcriptomes with those governed by *daf-9(yh1)* and *daf-9(-)* appear to suggest that the two *daf-9* alleles have the most differential impact on genes both positively and negatively regulated by PMK-1 signaling followed by SKN-1; effects on DAF-16 targets seem to be the same. PMK-1 is critical for *daf-2* mutants' immune resistance and directs multiple downstream transcription factors besides SKN-1 and DAF-16. It is very surprising then that *daf18* null appears to cause up-regulation of PMK-1 targets in *daf-2*. Wouldn't one expect it to suppress them? And if this is indication of toxic 'hyperactivation' then, the data would warrant greater exploration of PMK-1 modulation in this process, not just SKN-1. How does *pmk-1* null/RNAi impact the immunity of *daf-2;yh1*?

> We thank the reviewer's comment. To determine the role of PMK-1 in the immunity of *daf-2(e1370); daf-18(yh1) [daf-2(-); daf-18(yh1)]* animals, we performed PA14 resistance assay. We found that *daf-18(nr2037) [daf-18(-)]* suppressed the enhanced PA14 resistance in *daf-2(-)* worms in a *pmk-1*-dependent manner (Supplementary Fig. 10e). Although *daf-2(-); daf-18(-)* animals displayed increased PMK-1 activity compared with *daf-2(-)* or *daf-2(-); daf-18(yh1)* mutants, reduced DAF-16/FOXO activity in *daf-2(-); daf-18(-)* animals may have decreased the survival of the worms on PA14. These results are also consistent with a previous report showing that genetic inhibition of DAF-16, which acts downstream of DAF-18, increases PMK-1 target gene expression in *daf-2* mutants (Troemel et al., 2006). We also found that PMK-1 was required for the enhanced pathogen resistance of *daf-2(-)* and *daf-2(-); daf-18(yh1)* animals and for the

normal survival of wild-type worms upon PA14 infection (Supplementary Fig. 10e,f). This is consistent with the notion that loss-of-function mutations in *pmk-1*, the key regulator of *C. elegans* immunity against pathogenic bacteria, reduce innate immunity against PA14 in these animals. Overall, our data indicate that reducing the activity of PMK-1 can extend lifespan in *daf-2(-); daf-18(-)* animals while decreasing immunity in these animals. These data are also consistent with a previous report showing that lifespan and immunity can be regulated in opposite directions by one genetic factor (Amrit et al., 2019). Together, our new data suggest that hyper-activation of PMK-1 can cause harmful impacts on lifespan. We described these results in the manuscript as follows,

Figure 6 legends, page 52, line 951: “See Supplementary Figure 10e,f for the requirement of PMK-1 for the decreased PA14 susceptibility of *daf-2(-)* and *daf-2(-); daf-18(yh1)* worms and for the normal survival of WT on PA14.”

Supplementary Figure 10 legends, page 22, line 198: “**Supplementary Figure 10: Effects of *daf-16* RNAi, *skn-1(gf)*, *skn-1(-)*, and *pmk-1* RNAi on lifespan and pathogen resistance.** (a-c) Effects of *daf-16* RNAi [*daf-16(-)*] (n = 120 for each condition) (a), *skn-1(lax188)* [*skn-1(gf)*] (n ≥ 295 for each condition) (b), *skn-1(zj15)* [*skn-1(-)*] (n ≥ 270 for each condition) (c), and *pmk-1* RNAi [*pmk-1(-)*] (n = 120 for each condition) (d) on the lifespan of wild-type (WT) and *daf-2(e1370)* [*daf-2(-)*] worms. Please note that we used *skn-1(zj15)*, a point mutation that causes missplicing and reduces the mRNA levels of *skn-1*, because a strong *skn-1* mutant allele, *skn-1(mg570)*, causes sickness and short lifespan²⁸; indeed, we found that *mg570* substantially increased vulval rupture phenotypes in worms: 6% in WT and

22% in *daf-2(-); daf-18(nr2037)* [*daf-18(-)*] backgrounds. (e,f) The effects of *pmk-1* RNAi ($n \geq 108$ for each condition) on the survival of *daf-2(-); daf-18(yh1)* and *daf-2(-); daf-18(-)* (e), and WT and *daf-2(-)* worms (f) against PA14 infection. Although *daf-2(-); daf-18(-)* animals displayed increased PMK-1 activity compared with *daf-2(-)* or *daf-2(-); daf-18(yh1)* mutants shown in Fig. 5, reduced DAF-16/FOXO activity in *daf-2(-); daf-18(-)* animals may have decreased the survival of the worms on PA14.

These results are consistent with a previous report showing that genetic inhibition of DAF-16/FOXO, which acts downstream of DAF-18/PTEN, increases PMK-1 target gene expression in *daf-2* mutants¹⁴. In contrast to lifespan results shown in Fig. 6d, we found that PMK-1 was required for the enhanced pathogen resistance of *daf-2(-)* and *daf-2(-); daf-18(yh1)* animals and for the normal survival of WT upon PA14 infection. Our data indicate that reducing the activity of PMK-1 can extend lifespan in *daf-2(-); daf-18(-)* animals, while decreasing immunity in these animals. These data are consistent with a previous report showing that lifespan and immunity can be regulated in opposite directions by one genetic factor²⁹.”

What does its phosphorylation status look like in these animals vs. the double mutant with the null?

> Following the reviewer's comments, to address whether *daf-18(nr2037)* and *daf-18(yh1)* differentially affected the activity of PMK-1, we measured phospho-PMK-1 levels in wild-type N2, *daf-2(-)*, *daf-2(-); daf-18(yh1)*, and *daf-2(-); daf-18(nr2037)* animals by using western blot assays. We used *pmk-1(km25)* animals as a negative

control for antibody validation. We found that *daf-18(nr2037)* increased PMK-1 activity in *daf-2(-)* mutants, whereas *daf-18(yh1)* tended to have smaller effects on the increase in PMK-1 activity than *daf-18(-)* (Fig. 5j). Therefore, PMK-1 activity measured by phosphorylation status of PMK-1 appears to be higher in *daf-2(-); daf-18(nr2037)* animals than in *daf-2(-)* and *daf-2(-); daf-18(yh1)* mutants. These data are consistent with our model that hyperactivation of SKN-1 via PMK-1 is harmful for the longevity of *daf-2* mutants. We described these results in the manuscript as follows,

Results, Page 12, line 253: “We also found that the level of active, phospho-PMK-1 in *daf-2(-)* mutants was increased by *daf-18(-)*, while not being substantially affected by *daf-18(yh1)* (Fig. 5j).”

Figure 5 legends, page 50, line 938: “(j) Phospho-PMK-1 detection in WT, *daf-2(-)*, *daf-2(-); daf-18(yh1)*, and *daf-2(-); daf-18(-)* by using western blot assay (N = 5). *pmk-1(km25)* [*pmk-1(-)*] animals were used for the antibody validation (N = 2). α -tubulin was used as a loading control.”

Methods, page 37, line 779: “Western blot assay was performed as described previously with minor modifications⁷⁸. Briefly, bleached eggs were placed on *E. coli* OP50-seeded NGM plates. Hatched worms from the eggs were then allowed to develop to pre-fertile or day 1 young adults at 20°C, and were subsequently washed three times by using M9 buffer. The worms were then frozen in liquid nitrogen with 2X Laemmli sample buffer (#161-0747, Bio-Rad, Contra Costa County, CA, USA) containing 5% 2-mercaptoethanol (M3148, Sigma, St. Louis, MO, USA), boiled for 10 min at 98°C, and vortexed for 10 min. After centrifugation of the worm lysates at

15,871 g for 30 min, the supernatants were loaded to 10% SDS PAGE. The proteins were then transferred to PVDF membrane (#10600021, GE healthcare, Chicago, IL, USA) at 300 mA for 1 hr. The membranes were incubated with 5% bovine serum albumin solution in 1x TBS-T [24.7 mM Tris-HCl (pH 7.6), 137 mM NaCl, 2.7 mM KCl and 0.1% Tween 20] for blocking at room temperature for 30 min. Primary antibodies against phospho-PMK-1 (1:1,000, #9211, Cell Signaling Technology, Danvers, MA, USA), α -tubulin (1:2,000, sc-32293, Santa Cruz Biotechnology, Dallas, TX, USA), or His (1:1,000, #2365, Cell signaling technology, Danvers, MA, USA) were treated to the membranes overnight at 4°C. The membranes were then washed four times for 15 min using 1x TBS-T followed by incubating with secondary antibodies against rabbit (1:5,000, #SA8002, ABfrontier, Seoul, South Korea) or mouse (1:5,000, #SA8001, ABfrontier, Seoul, South Korea). After washing membranes with 1x TBS-T for 15 min, the membranes were then treated with ECL substrate (#1705061, Bio-Rad, Contra Costa County, CA, USA) for detecting protein bands. Images were visualized with ChemiDoc XRS⁺ system (Bio-Rad, Contra Costa County, CA, USA), and analyzed by using Image Lab software (Bio-Rad, Contra Costa County, CA, USA).”

Lastly, the ‘differential’ effects of daf-16 RNAi and skn-1(gof) on survival on pathogen of daf2;yh1 vs. daf-2>null (Fig. 5) cannot be used to draw the strong conclusions arrived at here. daf-16 RNAi suppressing daf-2; yh1 mutants long lifespan is taken as evidence that DAF-16 activity is maintained in daf-2 mutants. what about skn-1 rnaï/mutation?

> We thank the reviewer's comment that showing the impacts of *skn-1* loss of function on the lifespan of *daf-2(-); daf-18(nr2037)* animals would strengthen our points. We therefore performed lifespan and health span (motility) assays with a *skn-1* reduction-of-function mutation, *skn-1(zj15)* (Fig. 6c,g) [Please note that we used *skn-1(zj15)*, a point mutation that causes mis-splicing and reduces mRNA levels of *skn-1* (Tang et al., 2015), because a strong *skn-1* mutant allele, *skn-1(mg570)*, causes sickness and short lifespan (Lehrbach and Ruvkun, 2016); indeed, we found that *mg570* substantially increased vulval rupture phenotypes in worms: 6% in wild-type N2 and 22% in *daf-2(-); daf-18(nr2037)* backgrounds]. We found that *skn-1(zj15)* extended the short lifespan of *daf-2(e1370); daf-18(nr2037)* animals (Fig. 6c). In contrast, we found that *skn-1(zj15)* decreased the long lifespan of *daf-2(e1370); daf-18(yh1)* animals, indicating that retaining a proper SKN-1 activity in *daf-2(e1370); daf-18(yh1)* is required for longevity in these animals (Fig. 6c). These results suggest that long and healthy lifespan is sensitive to SKN-1 activity under reduced IIS conditions. We also measured motility (swimming) span of these worms but *skn-1(zj15)* did not affect either the motility of *daf-2(e1370); daf-18(nr2037)* or that of *daf-2(e1370); daf-18(yh1)* animals (Fig. 6g). The weak impact of *skn-1(zj15)* on the motility of worms may be due to a weak nature of the *zj15* allele, which may not affect the motility of worms in general. We included these results in the manuscript as follows,

Results, page 14, line 290: "Conversely, we found that a reduction-of-function allele *skn-1(zj15)* [*skn-1(-)*]⁴⁷ extended the short lifespan of *daf-2(-); daf-18(-)* animals, but not that of *daf-2(-); daf-18(yh1)* animals (Fig. 6c)."

Figure 6 legends, page 51, line 943: “(a-d) Effects of *daf-16(RNAi)* [*daf-16(-)*] (a), *skn-1(lax188)* [*skn-1(gf)*] (b), *skn-1(zj15)* [*skn-1(-)*] (c), and *pmk-1(RNAi)* [*pmk-1(-)*] (d) on the lifespan of *daf-2(e1370); daf-18(yh1)* [*daf-2(-); daf-18(yh1)*] and *daf-2(-); daf-18(nr2037)* [*daf-18(-)*] animals (n ≥ 240 for each condition). All the lifespan assays were performed at least twice. (e-h) Effects of *daf-16(-)* (e), *skn-1(gf)* (f), *skn-1(-)* (g), and *pmk-1(-)* (h) on the swimming rate (motility) of *daf-2(-); daf-18(yh1)* and *daf-2(-); daf-18(-)* animals at day 0 and day 7 adulthoods (n = 30 for each condition, from three independent trials). See Supplementary Figure 10c for the effects of *skn-1(-)* on *daf-2(-)* and wild-type (WT) animals. See Supplementary Figure 10e,f for the requirement of PMK-1 for the decreased PA14 susceptibility of *daf-2(-)* and *daf-2(-); daf-18(yh1)* worms and for the normal survival of WT on PA14. We found that *skn-1(-)* did not affect either the motility of *daf-2(-); daf-18(-)* or that of *daf-2(-); daf-18(yh1)* worms in g. The weak impact of *skn-1(-)* may be due to the weak nature of *zj15* allele, which needs to be tested in various other genetic backgrounds in future research.”

Supplementary Figure 10 legends, page 22, line 199: “(a-c) Effects of *daf-16 RNAi* [*daf-16(-)*] (n = 120 for each condition) (a), *skn-1(lax188)* [*skn-1(gf)*] (n ≥ 295 for each condition) (b), *skn-1(zj15)* [*skn-1(-)*] (n ≥ 270 for each condition) (c), and *pmk-1 RNAi* [*pmk-1(-)*] (n = 120 for each condition) (d) on the lifespan of wild-type (WT) and *daf-2(e1370)* [*daf-2(-)*] worms. Please note that we used *skn-1(zj15)*, a point mutation that causes mis-splicing and reduces mRNA levels of *skn-1*, because a strong *skn-1* mutant allele, *skn-1(mg570)*, causes sickness and short lifespan²⁸; indeed, we found that *mg570* substantially increased vulval rupture phenotypes in

worms: 6% in WT and 22% in *daf-2(-); daf-18(nr2037)* [*daf-18(-)*] backgrounds.”

3. The ‘transcriptomic analyses’ comparing *daf-2* gene expression changes governed by different factors (Figs 4 and S6) are described with little clarity in the results section and the phrasing is very confusing about genes positively or negatively regulated by a given factor. It is not explicit that this was harnessing data generated by previous studies. More importantly, the rationale for choosing the many genes is not elaborated. The presentation of the data in the figures can be improved as well. The most striking observation is the differential impact of the two *daf-9* alleles on ‘up’ and ‘down’ PMK-1 targets and then SKN-1 target. This can be shown in the main figures while the data with the remaining can be moved to the supplement.

> We appreciate the reviewer’s comments. We revised the manuscript for improving the clarity of the statements and added the rationale in the manuscript for choosing the genes used for RNA-seq. analysis in Fig. 5a and Supplementary Fig. 7 as follows,

Results, page 11, line 228: “For this analysis, we compared our RNA sequencing data to all the published transcriptome data that were obtained using animals with genetically inhibited *daf-2* (Fig. 5a, Supplementary Fig. 7, and Supplementary Table 6)^{23,34-44}.”

> We also added new cumulative plots showing the differential impact of *daf-18(yh1)* and *daf-18(nr2037)* on the expression of SKN-1 and PMK-1 targets in *daf-2* mutants for better understanding (Fig. 5b-e).

Figure 5 legends, page 49, line 914: “(b-e) Cumulative fraction of genes in an ascending order of the extent of gene expression changes conferred by *daf-18(yh1)* and *daf-18(-)* in *daf-2(-)* animals. (b,c) Shown are genes whose expression was upregulated (b) and downregulated (c) in *daf-2(-)* and *daf-2(e1368)* worms compared to *skn-1(zu67)* mutants²³. (d,e) Genes whose expression was upregulated (d) and downregulated (e) in *daf-2(e1368)* worms compared to *pmk-1(km25)* mutants are shown³⁵.”

4. In the manuscript (eg., lines 25, 64, 35) the authors use multiple assays as ‘fitness parameters’- development time, dauer entry, brood size, stress response, mobility etc.,. While technically this may be correct, it makes for confusing reading, especially since healthspan/fitness measures in the aging field usually refer to aspects of adult physiology. It may be helpful to distinguish developmental fitness parameters (development, dauer) from adult functionality metrics (mobility, immunity), especially in describing the impact of the *yh1* on them.

> We thank the reviewer’s comment. Following the reviewer’s suggestion, we revised the manuscript for clarity as follows,

Introduction, page 3, line 41: “In addition, the lifespan of wild nematode strains negatively correlates with growth rates², a key developmental fitness parameter.”

Introduction, page 4, line 70: “In this study, we aimed to uncouple the increased lifespan and decreased fitness, including developmental defects and decreased adult functionality metrics, exhibited by *daf-2* mutant *C. elegans*.”

Results, page 8, line 146: “Next, we assessed the developmental parameters and adult functionality metrics, including reproduction, motility, and feeding rates.”

Discussion, page 15, line 310: “Uncoupling the association between longevity and reduced fitness, including developmental defects and decreased adult functionality metrics, has been a major challenge in the field of aging research.”

5. The alleles *yh2* and *yh3* that map to *daf-16* appear to completely suppress *daf-2* immunity unlike *yh1* (Fig. 1b-d) and in any case are not investigated further in the article. This data can be removed/moved to the supplement and replaced with data showing the DAF-18::mCherry construct dauer/immunity rescue data (Fig. S2).

> We thank the reviewer’s comment. We moved the *yh2* and *yh3* survival data to the supplementary figures (Supplementary Fig. 1c,d) and replaced with the DAF-18::mCherry construct dauer/immunity rescue data (Fig. 1b,c).

Figure 1 legends, page 40, line 820: “(b,c) *mCherry::daf-18* fully rescued dauer formation ($n \geq 433$ for each condition, from two independent trials) (b) and the increased survival of worms on the pathogenic bacteria PA14 ($n \geq 180$ for each condition, with big lawn where worms do not have space for an avoidance behavior) (c) in *daf-2(-); daf-18(yh1)* mutants to the levels of *daf-2(-)* animals.”

Supplementary Figure 1 legends, page 2, line 8: “(b-d) Survival curves of *daf-2(-); daf-18(yh1)* (b), *daf-16(yh2); daf-2(-)* (c), and *daf-16(yh3); daf-2(-)* (d) mutants on PA14 compared with *daf-2(-)* mutant and wild-type (WT) worms ($n = 180$ for each condition). (e-h) Molecular natures of *yh1*, *yh2*, and *yh3* alleles identified from our

mutagenesis screen (See Supplementary Table 3 for details).”

6. Data on the syb499 mutants survival on PA14 is not shown in Fig. 1 as mentioned (line 107).

> We thank the reviewer’s comment. We included the missing information in the manuscript as follows,

Results, page 6, line 114: “Moreover, we confirmed that outcrossed *daf-2(-); daf-18(yh1)* and *daf-2(-); daf-18(syb499)* [a CRISPR/Cas9 knock-in allele that contains the same mutation as *daf-18(yh1)*] animals exhibited increased survival on PA14 infection and enhanced clearance of PA14 compared with wild-type animals (Fig. 1d-g and Supplementary Fig. 6c).”

7. Many of the supplementary tables (eg. S1-4) can be converted to .pdf versions/incorporated into the supplementary data file to make them easily accessible to the readers.

> We appreciate the reviewer’s comment. We converted the supplementary tables into PDF versions and included in the Supplementary Information file for better access to the readers. We also included the same supplementary tables as a spreadsheet file (Source Data) following the journal’s instruction, which recommend that the authors provide a single spreadsheet file with line graph data for each figure/table in a separate sheet.

8. The article requires some careful editing for nomenclature (eg. line 2: daf2/insulin/IGF1; lines 2, 22, 23: the mammalian homologs are FOXO3A and NRF2) and phrasing/sentence construction (eg., line 40: 'survival against ...stresses'; line 46: delete 'mutant'; line 104)

> We thank the reviewer's comment. We edited the designated nomenclature errors. Following are some examples of the changes we made.

We changed the title, line 2, as follows: "A *PTEN* variant uncouples longevity from impaired fitness in *Caenorhabditis elegans* with reduced insulin/IGF-1 signaling"

Introduction, page 3, line 46: "*daf-2* mutants also exhibit enhanced resistance to various stresses, such as oxidative, heat, osmotic, and pathogenic stresses⁵⁻⁷."

Introduction, page 3, line 52: "In addition, *daf-2* reduction-of-function alleles lead to reduced developmental rate, brood size, and motility, all of which are important biological attributes for competitive fitness in nature."

In addition, we changed all the "SKN-1/NRF" to "SKN-1/NRF2" in our manuscript. However, because *C. elegans* DAF-16 is a homolog of multiple FOXO proteins in mammals and therefore we retained DAF-16/FOXO.

References

Amrit, F.R.G., Naim, N., Ratnappan, R., Loose, J., Mason, C., Steenberge, L., McClendon, B.T., Wang, G., Driscoll, M., Yanowitz, J.L., *et al.* (2019). The longevity-promoting factor, TCER-1, widely represses stress resistance and innate immunity. *Nat Commun* 10, 3042.

Ayyadevara, S., Tazearslan, C., Bharill, P., Alla, R., Siegel, E., and Shmookler Reis, R.J. (2009). *Caenorhabditis elegans* PI3K mutants reveal novel genes underlying exceptional stress resistance and lifespan. *Aging Cell* 8, 706-725.

Ewald, C.Y., Landis, J.N., Porter Abate, J., Murphy, C.T., and Blackwell, T.K. (2015). Dauer-independent insulin/IGF-1-signalling implicates collagen remodelling in longevity. *Nature* 519, 97-101.

Friedman, D.B., and Johnson, T.E. (1988). A mutation in the *age-1* gene in *Caenorhabditis elegans* lengthens life and reduces hermaphrodite fertility. *Genetics* 118, 75-86.

Gems, D., Sutton, A.J., Sundermeyer, M.L., Albert, P.S., King, K.V., Edgley, M.L., Larsen, P.L., and Riddle, D.L. (1998). Two pleiotropic classes of *daf-2* mutation affect larval arrest, adult behavior, reproduction and longevity in *Caenorhabditis elegans*. *Genetics* 150, 129-155.

Lehrbach, N.J., and Ruvkun, G. (2016). Proteasome dysfunction triggers activation of SKN-1A/Nrf1 by the aspartic protease DDI-1. *Elife* 5.

Myers, M.P., Stolarov, J.P., Eng, C., Li, J., Wang, S.I., Wigler, M.H., Parsons, R., and Tonks, N.K. (1997). P-TEN, the tumor suppressor from human chromosome 10q23, is a dual-specificity phosphatase. *Proc Natl Acad Sci U S A* 94, 9052-9057.

Patel, D.S., Garza-Garcia, A., Nanji, M., McElwee, J.J., Ackerman, D., Driscoll, P.C., and Gems, D. (2008). Clustering of genetically defined allele classes in the *Caenorhabditis elegans* DAF-2 insulin/IGF-1 receptor. *Genetics* 178, 931-946.

Tang, L., Dodd, W., and Choe, K. (2015). Isolation of a Hypomorphic *skn-1* Allele That Does Not Require a Balancer for Maintenance. *G3 (Bethesda)* 6, 551-558.

Troemel, E.R., Chu, S.W., Reinke, V., Lee, S.S., Ausubel, F.M., and Kim, D.H. (2006). p38 MAPK regulates expression of immune response genes and contributes to longevity in *C. elegans*. *PLoS Genet* 2, e183.

Zhang, X.C., Piccini, A., Myers, M.P., Van Aelst, L., and Tonks, N.K. (2012). Functional analysis of the protein phosphatase activity of PTEN. *Biochem J* 444, 457-464.

REVIEWERS' COMMENTS

Reviewer #1 (Remarks to the Author):

The authors have made a very serious effort to take into consideration all the three reviewers' comments. This revised manuscript has grown substantively from the first submission, containing rich new experimental data that strengthened the main conclusions of the paper. Remarkably, the authors have now defined the mechanism underlying how *daf-18(yh1)* allele uncouples longevity from detrimental effects in reduced insulin signaling-mutant animals. Of notable interest is that the *daf-18(yh1)* allele substantially reduced lipid phosphatase activity while retaining protein phosphatase activity.

The authors have sufficiently addressed my concerns as well as the other Reviewers' concerns and I believe that the paper is appropriate for publication in Nature Communications.

Reviewer #2 (Remarks to the Author):

The authors have addressed all of my concerns and I was happy to see the additional data with more alleles and the protein tyrosine phosphatase activity of the PTEN C105Y variant. This new data makes the manuscript much stronger, but I still feel they have not entirely ruled out the "threshold model" for suppression of *daf-2* phenotypes. I suggest they change the following line:

"Despite the limitation due to the small number of tested *daf-18* mutant alleles, these data are not consistent with the possibility that the threshold for the suppression of constitutive dauer phenotype in *daf-2(-)* mutants is lower than that of enhanced pathogen resistance. "

to something along these lines:

"Due to the small number of tested *daf-18* mutant alleles, we cannot rule out different threshold levels for suppression of *daf-2* phenotypes, however, these data are not consistent with the possibility that the threshold for the suppression of constitutive dauer phenotype in *daf-2(-)* mutants is lower than that of enhanced pathogen resistance."

Reviewer #3 (Remarks to the Author):

the authors have addressed all my concerns with more-than-adequate experimentation and analyses. i am very happy to endorse publication. congratulations on an excellent study!
arjumand ghazi

We really appreciate having this opportunity to further revise our manuscript entitled, "A *PTEN* variant uncouples longevity from impaired fitness in *Caenorhabditis elegans* with reduced insulin/IGF-1 signaling". We thank all of the three reviewers for their very positive comments on our manuscript.

Our point-by-point responses to the reviewers' comments are described below. We hope you find this further revised manuscript is suitable for publication in *Nature Communications*.

Best regards,

Seung-Jae V. Lee (On behalf of all the authors)

Reviewer #1 (Remarks to the Author):

The authors have made a very serious effort to take into consideration all the three reviewers' comments. This revised manuscript has grown substantively from the first submission, containing rich new experimental data that strengthened the main conclusions of the paper. Remarkably, the authors have now defined the mechanism underlying how *daf-18(yh1)* allele uncouples longevity from detrimental effects in reduced insulin signaling-mutant animals. Of notable interest is that the *daf-18(yh1)*

allele substantially reduced lipid phosphatase activity while retaining protein phosphatase activity.

The authors have sufficiently addressed my concerns as well as the other Reviewers' concerns and I believe that the paper is appropriate for publication in Nature Communications.

> We appreciate the reviewer's comments regarding our efforts.

Reviewer #2 (Remarks to the Author):

The authors have addressed all of my concerns and I was happy to see the additional data with more alleles and the protein tyrosine phosphatase activity of the PTEN C105Y variant.

This new data makes the manuscript much stronger, but I still feel they have not entirely ruled out the "threshold model" for suppression of daf-2 phenotypes. I suggest they change the following line:

"Despite the limitation due to the small number of tested daf-18 mutant alleles, these data are not consistent with the possibility that the threshold for the suppression of constitutive dauer phenotype in daf-2(-) mutants is lower than that of enhanced pathogen resistance. "

to something along these lines:

"Due to the small number of tested daf-18 mutant alleles, we cannot rule out different

threshold levels for suppression of *daf-2* phenotypes, however, these data are not consistent with the possibility that the threshold for the suppression of constitutive dauer phenotype in *daf-2(-)* mutants is lower than that of enhanced pathogen resistance.”

> We appreciate the reviewer's insightful comment. We revised the text following the reviewer's suggestion,

Results, page 11, line 214: “Due to the small number of tested *daf-18* mutant alleles, we cannot rule out different threshold levels for suppression of *daf-2(-)* mutant phenotypes. However, these data are not consistent with the possibility that the threshold for the suppression of constitutive dauer phenotype in *daf-2(-)* mutants is lower than that of enhanced pathogen resistance.”

Reviewer #3 (Remarks to the Author):

the authors have addressed all my concerns with more-than-adequate experimentation and analyses. i am very happy to endorse publication. congratulations on an excellent study!

arjumand ghazi

> We appreciate the reviewer's very positive comments.